# Intelligent Operation and Maintenance of Wind Turbines Gearboxes via Digital Twin and Multi-Source Data Fusion

**DOI:** 10.3390/s25071972

**Published:** 2025-03-21

**Authors:** Tiantian Xu, Xuedong Zhang, Wenlei Sun, Binkai Wang

**Affiliations:** School of Intelligent Manufacturing Modern Industry, Xinjiang University, Urumqi 830046, China; xutiantian@stu.xju.edu.cn (T.X.); 107556522242@stu.xju.edu.cn (X.Z.); wangbk@stu.xju.edu.cn (B.W.)

**Keywords:** wind turbine, whale optimization algorithm, time convolutional network, intelligent O&M, fault early warning

## Abstract

Wind turbine operation and maintenance (O&M) faces significant challenges due to the complexity of equipment, harsh operating environments, and the difficulty of real-time fault prediction. Traditional methods often fail to provide timely and accurate warnings, leading to increased downtime and maintenance costs. To address these issues, this study systematically explores an intelligent operation and maintenance method for wind turbines, utilizing digital twin technology and multi-source data fusion. Specifically, it proposes a remote intelligent operation and maintenance (O&M) framework for wind turbines based on digital twin technology. Furthermore, an algorithm model for multi-source operational data analysis of wind turbines is designed, leveraging a Whale Optimization Algorithm-optimized Temporal Convolutional Network with an Attention mechanism (WOA-TCN-Attention). The WOA is used to optimize the hyperparameters of the TCN-Attention model. Then, the gearbox fault alarm threshold and warning threshold are set using the statistical characteristics of the residual values, and the absolute value of the residuals is used to determine the abnormal operating state of the gearbox. Finally, the proposed method was validated using operational data from a wind farm in Xinjiang. With input data from multiple sources, including seven key parameters such as temperature, pressure, and power, the method was evaluated based on EMAE, ERMSE, and EMAPE. The results demonstrated that the proposed method achieved the smallest prediction error and provided effective early warnings 18 h and 33 min prior to actual failures, enabling real-time and efficient operation and maintenance management for wind turbines.

## 1. Introduction

With the continuous increase in global energy demand and growing environmental awareness, wind energy, as a renewable and clean energy source, is rapidly developing and being widely adopted worldwide. According to the 72nd edition of the World Energy Statistics Yearbook 2023, renewable energy generation has increased by 14%, with total wind power generation reaching 2104.8 TWh. Of this, China’s wind power generation amounted to 762.7 TWh, ranking first globally. Meanwhile, the installed capacity of wind turbines has been steadily growing. Figure 1a illustrates the global total installed capacity of wind turbines from 2013 to 2022 (data source: GWEC), showing a year-on-year increase in capacity. Additionally, the Global Wind Energy Council (GWEC) Global Wind Report 2023 forecasts that between 2023 and 2027, the global wind power installed capacity will increase by 680 GW. As a major energy consumer, China’s wind turbine installed capacity is also expanding rapidly, as shown in Figure 1b. According to the GWEC 2024 Global Wind Report, in 2023, the global wind power installed capacity grew by 117 GW, marking a 50% increase compared to 2022. This brought the cumulative global wind power installed capacity to 1021 GW, surpassing the 1000 GW milestone, and reflecting a 13% growth from 2022. Notably, in 2023, global onshore wind power installations surpassed 100 GW for the first time, reaching 106 GW, which represents a 54% increase from 2022 and the highest increase on record. China continues to be one of the largest global markets for new onshore wind power installations. The rapid growth in the installed capacity of wind turbines has significantly increased the scale and operational complexity of the equipment, creating an urgent demand for more efficient maintenance strategies.

Due to the complex operating environment and variable working conditions of wind turbines, such as operating in humid, corrosive, sandy, and unstable wind conditions, wind turbines are highly susceptible to faults. Furthermore, in order to increase power generation, wind turbine manufacturers have extended the length of the blades from an initial 10 m to 107 m, and the tower height has increased from 50 m to 150 m [1]. This has resulted in increasingly larger wind turbines, which in turn has led to a higher incidence of faults. In actual operation, even minor faults in wind turbines can eventually lead to major accidents, such as blade damage, cabin fires, and tower collapses. Under traditional maintenance models, wind farms typically schedule regular maintenance and inspections, which inevitably incur high operation and maintenance costs and result in long downtimes [2,3]. Additionally, the wind farm environment is not conducive to the long-term presence of maintenance personnel, further increasing the operation and maintenance costs [4]. To ensure the stable operation of wind turbines and reduce maintenance costs, research into condition monitoring and fault prediction methods during their service life has become an inevitable trend. By real-time monitoring of the operational status of wind turbines, potential faults can be predicted in time, and maintenance personnel can be notified to intervene, thus preventing major accidents. This is of great significance for achieving the maximum economic benefits of wind turbines. Currently, the main monitoring and fault prediction methods for wind turbines include vibration monitoring, electrical signal monitoring, temperature monitoring, and oil/gas monitoring [5]. However, these existing methods face issues such as low intelligence, poor three-dimensional visualization, and long latency [6].

With the development of information technology, some researchers have proposed new monitoring and fault prediction methods that can more comprehensively monitor the operation status and fault prediction of wind turbines, such as mechanistic models, big data mining, and neural networks [7]. However, research has shown that these methods mainly perform fault prediction through offline analysis, which introduces high latency during the fault prediction process, leading to delayed problem detection and an inability to determine the real-time operating state of the wind turbine, thus affecting the monitoring efficiency of the wind turbine to some extent [8].

## 2. Related Works

Many scholars have conducted in-depth research and exploration on intelligent detection and operation and maintenance methods for wind turbines, achieving significant progress. For example, scholars such as Tian Xinying [9], Dong Liang, and others analyzed and studied onshore wind turbine systems, and in combination with the LabVIEW software platform developed and designed a wind turbine condition monitoring system. They approached the design from four aspects: data acquisition, data uploading, data storage, and data display [10]. Based on the main content and technical features of the IEC 61400-25 standard, a standardized information model for wind turbines and wind farms was developed. Additionally, a remote integrated monitoring system for offshore wind farms was developed, based on the .Net framework, jquery AJAX technology, and configuration technology. This system integrates data acquisition, monitoring and evaluation, and fault alarm functions. Zhang Chao [11] and others selected the average wind speed, average theoretical output power, and average actual output power from SCADA data under normal operating conditions of wind turbines to perform Johansen cointegration testing. They established a cointegration model and effectively monitored the operating status of the wind turbine using cointegration analysis. Zhao Yu [12] and Zhang Yuchen [13] selected the average wind speed, average theoretical output power, and average actual output power from SCADA data under normal operating conditions of wind turbines to perform Johansen cointegration testing. They established a cointegration model and effectively monitored the operating status of the wind turbine using cointegration analysis. Ziqi Wang [14] and others used multivariate state estimation techniques to construct a non-parametric, high-precision standard behavior model. They adapted the model through continuous learning methods, allowing for adaptive updates. The results showed that this approach achieved higher accuracy and lower false alarm rates. Jiayang Liu [15] and others used spatiotemporal graph neural networks to address the limitations in conventionality, interconnectivity, and locality of data processed by deep learning-based wind turbine state monitoring methods. They developed a wind turbine state monitoring method based on spatiotemporal graph networks. Pascal Dore [16] and others used fuzzy logic and the Fast-ESPRIT algorithm, related to the CAFH algorithm, to construct a real-time intelligent monitoring system for wind turbines based on fuzzy systems. This system addressed the economic losses caused by system braking due to false alarms in existing monitoring systems. Xiaohang Jin [17] and others utilized the large amount of data accumulated by the wind turbine’s SCADA system. They constructed a Mahalanobis space using historical data from the turbine’s normal operation and compared it with a trained behavior prediction model to detect anomalies in the wind turbine. Yueqi Wu [18] and others applied long short-term memory (LSTM) neural networks and the Kullback–Leibler divergence statistical tool for detecting and identifying gearbox-bearing faults. This approach efficiently enabled the monitoring of wind turbine operating conditions. Yang [19] and others proposed a monitoring method based on a deep joint variational autoencoder (JVAE). This method effectively used wind farm monitoring and data acquisition (SCADA) data to detect gearbox faults in wind turbines. Xiang [20] and others proposed a new wind turbine fault detection method based on an attention mechanism (AM), combining convolutional neural networks (CNN) and long short-term memory (LSTM) networks. This method improves the accuracy and effectiveness of detecting faults in wind turbines. Li Ming [21] and others proposed a gearbox fault prediction method based on extreme learning machines (ELM), which successfully enabled the early prediction of abnormal states in gearboxes. Zhu [22] proposed a combined method for real-time operational status prediction of wind turbines, aimed at reducing operation and maintenance costs and improving reliability. This method integrates long short-term memory (LSTM) networks with fuzzy comprehensive evaluation techniques. Cui [23] and others proposed a fault detection framework for wind turbine condition monitoring. This framework is capable of generating alerts for operational risks while reducing false alarms during normal operations. Wang Jian [24] and others proposed an oil temperature prediction model based on an improved particle swarm optimization (imPSO) and BP neural network. This model can identify potential gearbox faults earlier and demonstrates higher prediction accuracy compared to the standard PSO-BP model. Zhang [25] and others proposed an anomaly detection and diagnosis method for wind turbines based on LSTM-SDAE and XGBoost. They effectively validated the proposed method using SCADA data, demonstrating its strong performance in detecting and diagnosing anomalies in wind turbines. Yetis [26] proposed a new method for accurately predicting wind speed based on machine learning algorithms, which offers higher accuracy compared to other models. Wang [27] studied a gearbox health monitoring method for wind turbines based on lubrication oil pressure data and proposed a monitoring framework based on deep neural networks (DNNs), which can effectively identify the operation and condition of the gearbox, as well as impending faults. Yan [28] proposes a novel coarse-to-fine dual-scale time-frequency attention fusion network (CDTFAFN) for machinery fault diagnosis, which not only adequately considers the complementary information fusion of vibro-acoustic signal but also has robust feature learning capabilities in a noisy scenario. It has significant advantages in mechanical fault diagnosis. Ye [29] noted that existing diagnostic methods, which rely on fault information provided by a single sensor, are limited and incomplete, making it difficult to meet the need for accurate and reliable fault diagnosis in complex scenarios. To address these challenges, he proposed a Multi-Sensor Residual Convolutional Fusion Network (MRCFN) for intelligent bearing fault diagnosis, which achieves the complementarity and calibration of multi-sensor feature information, leading to high-precision diagnosis. Wang [30] et al. implemented rolling bearing fault signal classification using an optimized SVM model, and by applying dimensionality reduction techniques, they achieved high classification accuracy. Yin [31] et al. optimized the classification layer of a Long Short-Term Memory (LSTM) model using a cosine loss function and achieved fault classification for wind turbine gearboxes. Compared to previous improvements, this approach enhanced fault recognition accuracy. Hao [32] combined the 1D-CNN model with the LSTM model to construct a bearing fault classification model, which achieved relatively stable classification accuracy across datasets with different loads and signal-to-noise ratios, demonstrating good generalization ability. Liu [33] fused the statistical features and hidden features from multi-source signals and, based on this, constructed a model for rotating machinery fault diagnosis, which demonstrated high accuracy. Wang [34] proposed a multi-dimensional serial CNN model with an attention mechanism to extract and fuse feature information from multi-modal bearing signals. This approach implemented a third-order feature fusion strategy, improving the accuracy of the fault classification model.

In recent years, with the rapid development of industrial Internet technologies, the application of digital twin technology in industrial equipment has seen significant growth. Manuel Chiachío [35] utilized Internet technology to achieve real-time mapping of physical entities in virtual space. Furthermore, many scholars have constructed digital twin models, enabling real-time monitoring and visualization of large-scale equipment. These models provide an accurate and dynamic representation of equipment, facilitating enhanced operational management, predictive maintenance, and performance optimization [36,37,38,39]. At the same time, digital twin technology has begun to be applied in the field of wind power equipment. Tehrani K [40] developed a smart multiphysics approach for wind turbine design utilizing Industry 5.0, created and optimized a new blade profile using the non-dominated sorting genetic algorithm II (NSGA-II) for shape design, and proposed a 3D modeling of wind turbines. Qin Shengqiong [41] and others, in 2021, were among the first to explore the development trends and application prospects of digital twin technology in the innovative design of complex wind turbine systems (data source: China National Knowledge Infrastructure). Wang Xin [42] and others, using emerging digital twin technology, constructed a digital twin of the power grid based on a five-dimensional digital twin model. They summarized the challenges of applying digital twin technology to the power grid from six perspectives. Fang Fang [43] proposed the concept of information–physical real-time mapping and built a digital twin system for wind turbines. This system utilized existing SCADA data, but SCADA data mainly consist of low-frequency data, which are not sufficient to comprehensively monitor the operational status of wind turbines. Dongpeng Su [44], addressing the issue of delayed operation and maintenance (O&M) management for wind turbines, proposed a digital twin system model for wind turbines based on systems engineering theory. The model integrated simulation models, digital end models, and O&M management models to construct a wind turbine O&M digital twin prototype system, and its feasibility was verified. Xuanhui Zhao [45] designed and developed a cloud-based wind farm digital twin software system based on three components: host computer software, data service platform, and application examples. This system achieved the visualization of the wind turbine 3D model and data monitoring functionality. Xiang Zhao [46] developed a digital twin model for the drivetrain system of offshore wind turbines using torque dynamics models, online measurements, and fatigue damage assessment. He also proposed a state monitoring method based on the torque dynamics model. However, due to the coupled operating environment of wind turbines, the state monitoring is not precise enough, and others constructed a digital twin model of wind turbine parameter variations using component reduced-order modeling technology. This model can provide real-time predictions of structural responses and structural health conditions caused by wind-wave loads. However, this method did not account for the impact of real-time data integration into the model and other potential influences that may arise in practical applications. Obafemi O. Olatunji [47] pointed out that digital twin technology can predict failures of individual wind turbine components, making the monitoring and maintenance of wind turbines more efficient. However, its functionality is too simple. Montaser Mahmoud [48] and others built a digital twin system for wind turbines using physical systems, digital systems, connection systems, and service systems. This significantly improved the operation and maintenance efficiency of wind turbines, resulting in longer service life, shorter downtime, and higher safety performance. Shu Liu [49] and others constructed a digital twin model to predict wind power in real time and with high accuracy. They also proposed an ultra-short-term wind power prediction method based on digital twin technology, using a BP neural network to obtain the predicted values. In addition, digital twin technology in the field of wind turbine applications also covers areas such as structural response [50], life prediction [51], and reliability analysis [52].

The above studies have made significant progress in the intelligent monitoring and operation and maintenance of wind turbines. Data collection and monitoring technologies have gradually matured, and real-time monitoring of key operating parameters, such as temperature, vibration, and rotational speed, using sensors has laid a solid foundation for the operation monitoring and health management of wind turbines. However, existing monitoring methods have low visualization and poor real-time performance. Digital twin technology can effectively address these shortcomings. By creating a virtual model of the wind turbine through digital twins and connecting it in real time with physical equipment, sensor data from the field (such as temperature, vibration, rotational speed, current, etc.) can be transmitted in real time to the virtual model. Digital twins not only allow for the real-time representation of the wind turbine’s status but also enable the complete simulation of the turbine’s operation in a virtual space. This allows maintenance personnel to visually observe the dynamic changes in various parameters of the turbine and perform comprehensive monitoring. Despite a large number of studies applying machine learning and deep learning algorithms to mine and processing operational data from wind turbines, achieving fault feature recognition and diagnosis to some extent, the diverse operational parameters of wind turbines pose challenges. Traditional machine learning algorithms and convolutional neural networks (CNNs) have limited ability to fuse and analyze multi-source data. Moreover, they still face certain limitations when handling time-series data. Although models like Long Short-Term Memory (LSTM) networks can capture long-term dependencies in sequential data more effectively than traditional Recurrent Neural Networks (RNNs), they still face issues such as the gradient problem and long training times. Therefore, to address the real-time monitoring and intelligent fault warning issues of wind turbines and achieve efficient intelligent operation and maintenance, this study proposes an intelligent operation and maintenance framework for wind turbines based on digital twins and multi-source data fusion. The framework connects and maps the digital twin model of the wind turbine with the operational status of the physical entity in real time, and performs intelligent and efficient data analysis and fault warning based on machine learning and deep learning models. To further enhance the accuracy and time-series learning ability of the fault warning algorithm model, a causal dilated convolution and residual connection network structure based on a Temporal Convolutional Network (TCN) was designed. Multiple variable inputs were used to predict the gearbox oil sump temperature. The Whale Optimization Algorithm was employed to optimize the hyperparameters of the TCN-Attention model, improving the model’s prediction accuracy. The algorithm model was then deeply integrated with the digital twin model, thereby constructing a digital twin system for the intelligent operation and maintenance of wind turbines.

## 3. Materials and Methods

### 3.1. The Basic Structure of a Wind Turbine

A wind turbine, as a device that converts wind energy into electrical energy, operates on the principle of using wind as its power source. When wind blows across the turbine’s blades, the blades start to rotate under the force of the wind, thereby converting wind energy into mechanical energy. This mechanical energy is then transmitted to the generator through the wind turbine’s internal transmission system (gearbox), converting it into electrical energy. Taking the most typical Horizontal Axis Wind Turbine (HAWT) as an example, the overall structure of the wind turbine [53] is shown in Figure 2, and it typically consists of the wind rotor, nacelle, and tower. The wind rotor is the device that captures wind energy, and it usually includes components such as the blades, hub, nacelle cover, and pitch mechanism. Using aerodynamic principles, the blades convert wind energy into mechanical energy, causing the wind rotor to spin and thereby driving the main shaft of the wind turbine. The nacelle is the installation site for the key equipment of the wind turbine, where devices such as the gearbox, main shaft, generator, and yaw system are installed. Workers can access the nacelle via the tower to inspect and maintain the wind turbine. The tower is typically used to support the nacelle and the wind rotor. Terrain roughness, meteorological conditions (such as wind direction, wind speed distribution, turbulence intensity, etc.), and economic factors (such as tower manufacturing costs, installation difficulty, etc.) are all important factors that determine the optimal tower height. Generally, the taller the tower, the greater the wind speed the rotor can capture. Additionally, the tower is equipped with an elevator to facilitate worker access to the top. The gearbox, as the most critical component of the wind turbine drivetrain, is characterized by its complex structure, high failure rate, and significant maintenance challenges. Consequently, monitoring the operational status of the gearbox plays a crucial role in ensuring the reliability and efficiency of wind turbine operation and maintenance.

### 3.2. Remote Intelligent Operation and Maintenance Method for Wind Turbines Based on Digital Twin

The operating environment of wind turbines is highly complex and uncertain. External factors such as wind speed, wind direction, and temperature, as well as the structure and performance of the turbine itself, directly impact the turbine’s operating status. However, traditional monitoring methods and data analysis approaches often fail to meet the needs for accurate monitoring and optimization. By using digital twin technology, a virtual model synchronized with the actual physical turbine can be created, allowing real-time collection and analysis of various data from the turbine, accurately reflecting its operational status, and helping managers achieve more efficient scheduling and maintenance. Therefore, based on a thorough reference to the energy Internet digital twin system framework and the digital twin five-dimensional model theory [54], this paper proposes a method for building a digital twin system for intelligent monitoring and operation of wind turbines. Through the fusion and analysis of multi-dimensional data, the digital twin system can accurately monitor and predict the health status of the wind turbine, detect potential issues in time, and provide early warnings before problems occur, thus avoiding downtime losses caused by failures and improving the reliability and economic performance of the wind turbine. As shown in Figure 3, this intelligent operation and maintenance digital twin system for wind turbines consists of four layers: the physical layer, data layer, model layer, and service layer.

#### 3.2.1. Physical Layer

The physical layer primarily includes wind turbine equipment, sensors, and other physical entities, which use IoT technology to monitor various operating parameters and environmental data of the wind turbine in real time. To comprehensively monitor the operational status of wind turbines, multiple sensors are deployed to collect key data, such as operational parameters, environmental information, grid status, rotation speed, temperature, vibration, set parameters, time information, and blade pitch angles. To meet the real-time processing requirements of massive data and reduce cloud data redundancy, data collection terminals utilize intelligent gateways and controllers with edge computing capabilities to preprocess data at the edge. At the edge, raw collected data are cleaned, processed, preprocessed, and stored to ensure high data quality and low redundancy when transmitted to the cloud. The cloud receives and stores the processed data from the edge, utilizing data sharing, analysis, and mining technologies, combined with mechanism models and algorithms, to provide real-time monitoring, anomaly warning, fault diagnosis, and operational maintenance services for wind turbines.

#### 3.2.2. Data Layer

The data layer consists of three parts: data transmission, data processing, and data center. The data transmission part involves communication protocols and transmission methods. The intelligent gateway or controller communicates with various sensors using the Modbus protocol to collect operational data from the wind turbines. Furthermore, data transmission employs protocols like TCP/IP and UDP to transfer data from edge devices to cloud servers. The data processing part involves a series of processing steps at the edge, such as format conversion, correlation analysis, data processing, cleaning, and anomaly data handling. The processed data are then transmitted via network protocols (e.g., TCP/IP, UDP) to the cloud server’s data center. On the cloud, parsing programs convert the received data into a standardized format and store them in a database for further use and analysis. The data center includes twin data of the wind turbine’s physical layer equipment, association rules, and other critical data resources. These are stored and managed using relational databases (e.g., MySQL, SQL Server, PostgreSQL) or non-relational databases (e.g., HBase, MongoDB, Redis), ensuring efficient storage and fast retrieval of data to support various business needs.

#### 3.2.3. Model Layer

The model layer mainly includes the three-dimensional digital twin model and mechanism models of the wind turbine. The digital twin model is a digital mirror of the physical entity, with functions like assembly constraints, simulated motion, and dynamic interaction between parts. This model can be created using various tools such as CAD modeling software, Unity, Revit, 3D rapid scanners, and it includes detailed assembly constraints, hierarchical relationships, etc. With lightweight technologies for 3D models (e.g., WebGL, Three.js), these models can be browsed and displayed in web applications, enabling cross-platform visualization. The mechanism model involves the working principles of wind turbines and various physical and mathematical models, including the HHT (Hilbert–Huang Transform) model, the convolutional neural network (CNN) model, and the bearing fault diagnosis model combining HHT and CNN. These models help to deeply understand the dynamic behaviors during turbine operation, and predict and analyze potential faults, especially in fault diagnosis. By combining the digital twin and mechanism models, the model layer provides powerful data support and analytical tools to the service layer, supporting applications such as 3D visual monitoring and bearing fault diagnosis. This improves the operational efficiency, maintenance accuracy, and fault warning capabilities of wind turbines. The model layer not only enables precise monitoring and simulation of wind turbines but also provides a scientific basis for fault prediction and intelligent diagnostics, promoting the intelligent operation and maintenance of wind power systems.

#### 3.2.4. Service Layer

The service layer primarily includes web applications for wind turbine management, fault diagnosis, anomaly warning, and decision making. This layer employs a range of front-end and back-end technologies, such as Vue, HTML5, CSS, JavaScript, WebGL, Unity, C#, MQTT, etc., to provide efficient and intuitive user interfaces along with robust backend support. One of the core functions of the service layer is real-time monitoring and fault diagnosis based on twin data. By integrating deep learning-based fault diagnosis models, the system can detect the wind turbine’s operating status in real time, identify potential faults, issue early warnings, and provide fault localization. This significantly improves the operational efficiency of the turbines and reduces downtime. Additionally, the service layer includes comprehensive management of wind turbines, supporting functions such as equipment status monitoring, operational data analysis, and anomaly event warning, helping users make effective operational decisions in a timely manner. Through WebGL and Unity technologies, users can intuitively view the operational status and condition of turbines via a 3D visualization interface, further enhancing user experience and operational convenience. The service layer combines advanced data analysis, visualization technologies, and intelligent diagnostic models, improving wind turbine management efficiency and providing intelligent decision support for operators.

### 3.3. Multi-Source Data Fusion Early Warning Model for Wind Turbines Based on WOA-TCN-Attention

This paper designs a multi-source wind turbine unit operation data analysis model, which combines the Whale Optimization Algorithm (WOA) with Time Convolutional Networks (TCNs) and Attention Mechanism, aiming to improve the fault diagnosis and early warning accuracy of wind turbine units. The model mainly consists of three parts: a deep learning architecture based on time convolutional networks, a feature weighting module incorporating the attention mechanism, and the whale optimization algorithm for hyperparameter optimization.

Time Convolutional Networks (TCNs) are deep learning models that are particularly suitable for processing time-series data, effectively capturing long-term dependencies within the data. Wind turbine operation data typically contain multiple time series, which have complex interrelationships. TCN can use its extended convolution structure to capture these relationships by utilizing multi-level convolution kernels, thereby enhancing the ability to extract temporal features from the data.

To further improve the model’s performance, the attention mechanism is introduced. The core idea of this mechanism is to dynamically adjust the weights of different input features, prioritizing those factors that have a greater impact on the prediction results. In this study, the temperature variation of the gearbox oil pool is influenced by multiple factors such as external environment and equipment operating conditions. Traditional models often fail to effectively identify the importance of these factors. By using the attention mechanism, the model can automatically assign appropriate weights to different factors, thus improving the accuracy of predictions.

Next, the Whale Optimization Algorithm (WOA) is used to optimize the hyperparameters of the TCN and Attention models. The selection of hyperparameters is critical to the performance of deep learning models, and traditional manual tuning methods are time-consuming and ineffective. WOA is a heuristic optimization algorithm that simulates the hunting behavior of whales, enabling it to find the optimal hyperparameter combination in a complex search space. Through WOA optimization, the model can more efficiently find the most suitable network structure and training parameters for the current problem, thereby improving the overall performance of the model.

Finally, a residual analysis-based method is used to set the fault alarm threshold for the gearbox. During model training, the statistical characteristics of the residuals are utilized to determine the deviation between the predicted and actual values. When the absolute value of the residual exceeds a preset threshold, it indicates a potential anomaly in the system, triggering the fault alarm mechanism. This method not only allows real-time monitoring of the gearbox’s operating status but also enhances the accuracy of fault prediction, providing early warnings for potential equipment failures, reducing downtime, and minimizing maintenance costs.

The WOA-TCN-Attention model, by combining the temporal feature extraction capabilities of deep learning, the adaptive weighting power of the attention mechanism, and the hyperparameter tuning ability of the Whale Optimization Algorithm, can effectively identify abnormal conditions of the gearbox in the complex multi-source data environment of wind turbine units, providing reliable technical support for fault prediction and maintenance decision making.

#### 3.3.1. Time Convolutional Network (TCN)

The Temporal Convolutional Network (TCN) is a deep learning architecture that combines one-dimensional convolutional neural networks with causal convolutions, specifically designed for processing time-series data. TCN addresses the limitations of traditional convolutional neural networks in handling time-series data through the use of causal dilated convolutions and residual connections. Causal dilated convolutions exponentially expand the receptive field of the model by introducing a dilation factor, enabling the model to capture long-term dependencies in time-series data while preserving its temporal structure. This design ensures causality in predictions, meaning that the output at the current time step depends only on the inputs at the current and previous time steps, thereby preventing information leakage from future time steps. For example, convolutional layers with dilation factors of 1, 2, and 4 can capture short-term, medium-term, and long-term dependencies, respectively. Additionally, TCN employs residual connections to directly pass input features to the output layer, mitigating the gradient explosion problem in deep networks and improving training stability. Residual connections not only accelerate convergence but also enhance the model’s ability to model complex temporal patterns. Figure 4 illustrates the detailed structure of causal dilated convolutions and residual connections. Causal dilated convolutions progressively expand the receptive field through hierarchical stacking and adjustment of the dilation factor, while residual connections facilitate effective gradient propagation by transmitting information across layers. These design features enable TCN to excel in processing long-sequence data while significantly improving prediction accuracy.

TCN uses one-dimensional dilated convolution for one-dimensional time series, $x_{n}=\left\{x_{t-w+1},\ldots ,x_{t-1,}x_{t}\right\}$, $x\in R^{n}$, convolution kernel $f:\left\{0,\cdots ,k-1\right\}\to R$, and its computational expression is Equation (1):(1)$$F(s)=(x*df)(s)=\sum_{i=0}^{k-1}f(i)\times x_{s-d\cdot i}$$

In Equation (1): d is the dilation factor; k is the size of the convolution kernel.

The residual connection module in the TCN network effectively avoids issues like gradient explosion and network degradation in deep traditional neural networks. The structure of the residual network module is shown in Figure 5. Residual connections can effectively train deep networks; they not only enable cross-layer information flow within the network but also allow the model to capture as much information as possible during feature extraction, thereby improving the accuracy of the model.

#### 3.3.2. Attention Mechanism

The idea of the attention mechanism is to simulate the human brain’s ability to focus on important information by filtering out key features from the input data and assigning them higher weights, while ignoring less relevant information. This enables the mechanism to effectively capture critical patterns in the input data and enhance the predictive performance of the model. Specifically, the attention mechanism first evaluates the relevance between each feature vector (e.g., temperature, pressure, power) in the input sequence and a target query vector. This relevance is measured using a metric called the “similarity score”, where a higher score indicates a greater influence of the feature on the target. Next, the similarity scores are normalized (e.g., using the Softmax function) to generate attention weights, which reflect the importance of each feature in the final output. Features with higher scores are assigned larger weights. Finally, the attention mechanism computes a weighted sum of all feature vectors based on their corresponding weights to produce the final output. This process ensures that key features contribute more significantly to the output, while the influence of less important features is minimized. The attention mechanism dynamically adjusts the weight distribution, allowing it to better capture critical information in the input data. For example, in the operational data of wind turbines, temperature variations may be assigned higher weights because they have a more significant impact on the gearbox oil sump temperature. This dynamic weight allocation mechanism not only enhances the model’s ability to capture key features but also improves prediction accuracy and robustness. This process primarily involves assigning greater weights to the key information, which is achieved through the calculation of weight coefficients. The formula for calculating the weight value is:(2)$$e_{t}=v\tanh(wh_{t}+b)$$(3)$$a_{t}=soft\max e_{t}$$(4)$$y_{t}^{\prime }=\sum_{t=1}^{n}a_{t}h_{t}$$

In Equations (2)–(4): $h_{t}$ is the output of the hidden layer in the TCN model; $e_{t}$ is $h_{t}$ the corresponding weight value; w and v are the weight parameter; b is the bias coefficient; $a_{t}$ is $h_{t}$ the corresponding weight value; and $y_{t}^{\prime }$ is the predicted value.

The article uses multiple variables as inputs to predict the temperature of the gearbox oil pool, with different variables having varying levels of impact on the temperature. The attention mechanism is employed to assign higher weights to key factors that have a significant effect on the temperature change, thereby improving the accuracy of the temperature prediction.

#### 3.3.3. TCN-Attention Model

The core idea of the TCN-Attention model is to introduce the attention mechanism based on the TCN (Temporal Convolutional Network), by assigning different weights to the input data at different time steps. This helps the model focus more on important time points and features. This mechanism effectively enhances the model’s ability to capture temporal dependencies in the data and, in turn, improves prediction accuracy. The TCN-Attention model consists of an input layer, TCN network layer, attention mechanism, and output layer.

First, the input data are fed into the input layer. For example, multi-source data such as the temperature at the front end of the gearbox high-speed shaft, the temperature at the rear end of the gearbox high-speed shaft, the inlet oil temperature of the gearbox, the active power of the generator, the cooling water temperature of the gearbox, the inlet pressure of the gearbox, and the oil sump temperature of the gearbox are used as inputs. Then, the TCN network layer processes the data to extract a basic feature set. The attention layer then assigns weights to distinguish between important and less important features. Finally, the fully connected layer generates the final prediction result. The input data consist of time-series data used to predict the gearbox oil temperature, which combine causal convolutions and dilated convolutions for feature extraction. ReLU activation functions are used between hidden layers to introduce nonlinear transformations, and residual connections are introduced between adjacent hidden layers. After the hidden layers’ output, the attention mechanism automatically calculates the influence of each time step’s feature information (h*ₜ*) on the temperature change of the gearbox oil pool. Different weights are assigned to the feature information at each time step. The output layer is a fully connected layer that receives the weighted output vector from the attention mechanism and uses it as the predicted value for the gearbox oil temperature. The structure of the TCN-Attention model is shown in Figure 6.

#### 3.3.4. Parameter Optimization Based on WOA

The Whale Optimization Algorithm (WOA) [13] is a metaheuristic swarm intelligence algorithm that simulates the hunting mechanism of whales (bubble-net feeding method). Compared to traditional swarm intelligence algorithms, it has the advantages of simple computation steps, fast iteration speed, and the ability to avoid local optima [14,15,16]. This algorithm primarily consists of three stages: global search for prey, encircling the prey, and bubble-net attack. It can intelligently search in multi-dimensional space by adjusting the position vector of the particles, selecting the optimal solution from several candidate solutions based on the objective function. The basic principle of the Whale Optimization Algorithm is as follows.

The Whale Optimization Algorithm assumes that the position $X^{*}(t)$ of the target prey is the optimal solution. All whale individuals continuously iterate and update their relative positions X(t). The mathematical model is expressed as follows:(5)$$D=|C\cdot X^{*}(t)-X(t)|$$(6)X(t+1)=X(t)−A·D

In the Equations (5) and (6), t represents the iteration count; A and C are the control parameter, composed of a random real number and a decreasing factor, and D is distance between the whale individual and the prey.

In the Whale Optimization Algorithm (WOA), the Bubble Net Attack consists of the Constriction Encircling Mechanism and the Spiral Updating Mechanism. The Constriction Encircling Mechanism is shown in Equation (6), while the Spiral Updating Mechanism is shown in Equation (7). l is a (−1, 1) random number, and b is constant, representing the logarithmic spiral shape.(7)$$X(t+1)=X^{*}(t)+D_{q}e^{bl}\cos\theta (2\pi l)$$

Assuming that the whale selects either the constriction encircling mechanism or the spiral mechanism to update its position with a 50% probability each while capturing the target prey, the mathematical model can be expressed as Equation (8):(8)$$X(t+1)=\left\{\begin{array}{l} X(t)-A\cdot D,\;\;p<0.5\\ X^{*}(t)+D_{q}e^{bl}\cos\theta (2\pi l),\;p\geq 0.5 \end{array}\right.$$

An individual whale *m* is capable of randomly searching for targets within a given range, $X_{m}(t)$ where the whale’s position is randomly selected. The mathematical model is as follows in Equations (9) and (10):(9)$$D=|C\cdot X_{m}(t)-X(t)|$$(10)$$X(t+1)=X_{m}(t)-A\cdot D$$

## 4. Results

### 4.1. Model Training and Validation

This study uses the 2 MW #2 wind turbine of a wind farm in Xinjiang as an example; the wind turbine model uses the PEAB 4435.2 gearbox, with a power rating of 2200 kW, a maximum input speed of 16.7 r/min, and a rated transmission ratio of 1:100.6. Research institutions both domestically and internationally have analyzed a large amount of fault data from the operation of wind turbines through long-term records and statistics, aiming to identify the fault types that cause the most significant downtime and economic losses. The Spanish company EHN conducted a statistical analysis of fault data from 69 wind farms. The results showed that most wind turbine faults were concentrated in the gearbox, generator, and blades. Among these, gearbox faults accounted for the largest proportion of all fault types, ranging from 48% to 60%. Generator faults ranked second, accounting for 21% to 29%, while blade faults accounted for 11% to 17%. Gearbox failures resulted in prolonged downtime and significant economic losses. Therefore, real-time monitoring and early warning of the wind turbine gearbox have become particularly critical. To collect and measure the multi-source operational data of the gearbox, temperature sensors are placed at various locations, including the front and rear ends of the high-speed shaft, the oil inlet, the oil sump, and the inlet and outlet of the cooling water. Additionally, pressure sensors are installed at the oil inlet or outlet of the gearbox to ensure that the working pressure of the gearbox is accurately reflected. In addition, the SCADA system is used to monitor the generator power parameters of the wind turbine. These multi-source data are combined into a feature parameter vector representing the operational status of the wind turbine gearbox, which is then saved and analyzed in real time. The data sampling period is set to 1 min, and a total of 30,912 datasets were collected. After removing anomalies and handling missing values in the recorded operational data, a final set of 22,085 valid data points was obtained. In the operational data of the wind turbine, various parameters can affect the normal operation of the wind turbine. To evaluate the correlation between each variable and the predicted value, the Maximal Information Coefficient (MIC) is used to capture the nonlinear relationships between variables, with a relatively simple computation method. Therefore, the Maximal Information Coefficient (MIC) method [17] is employed, with the gearbox oil tank temperature as the target variable, to analyze the correlation between the gearbox variables and the predicted values, based on the calculation principle as shown in Equation (11).(11)$$MIC(X;Y)=\underset{a\times b<B}{\max}\frac{I(X;Y)}{\log_{2}(\min(a,b))}$$

The variables selected using the MIC method are used as the variable set for training the model. The correlation analysis results between the gearbox variables and the gearbox oil temperature are shown in Table 1.

The dimensional differences of various features may affect the prediction performance of the model. Therefore, the input data are normalized, and the calculation method is shown in Equation (12).(12)$$y_{i}=\frac{x_{i}-x_{\min}}{x_{\max}-x_{\min}}$$

In the Equation (12): $y_{i}$ represents the i -th normalized data; $x_{i}$ represents the i -th data before normalization; and $x_{\max}$, $x_{\min}$ are the maximum and minimum values of the variable x, respectively.

After normalization, 1000 data samples are selected as the test set, followed by 1400 data samples from the 24 h before the fault as the fault set, with the remaining data from normal operating conditions used as the training set. The PyTorch framework is used for training, and the prediction performance is compared among the LSTM model, TCN model, and TCN-Attention model. Seven feature parameters, including gearbox front-end temperature, gearbox rear-end temperature, gearbox inlet oil temperature, generator active power, gearbox cooling water temperature, gearbox inlet pressure, and gearbox oil tank temperature, are used as input data for the models. The models, including the WOA-optimized TCN-Attention model, LSTM model, TCN model, and TCN-Attention model, are trained. In this study, the Whale Optimization Algorithm (WOA) is employed, with an initial whale population of 30 and a maximum of 50 iterations; the optimal model parameter configuration scheme was found with the objective function of achieving the highest prediction accuracy of the model. The selection mainly focused on parameters such as learning rate, filters, convolution kernels, and dilation list, with the results as shown in Table 2. The Adam optimizer is used, with a batch size of 32 and 200 training epochs. The resulting prediction performance of the WOA-TCN-Attention model is shown in Figure 7. To make the curve more distinguishable, a sample range from 500 to 600 is selected to plot the predicted results of this model, as shown in Figure 8. The residuals of the prediction model are shown in Figure 9.

As shown in Figure 7, the prediction values of the WOA-TCN-Attention model are very close to the actual values, demonstrating excellent prediction performance. A comparison of the prediction performance of this model with the LSTM model, TCN model, and TCN-Attention model is made. To precisely assess the prediction performance of the above models, the evaluation metrics used are Mean Absolute Error (EMAE), Root Mean Square Error (ERMSE), and Mean Absolute Percentage Error (EMAPE). The evaluation metrics are given by Equations (13)–(15).(13)$$E_{MAE}=\frac{1}{n}\sum_{i=1}^{n}|(y_{i}-\overset{\wedge }{y_{i}})|$$(14)$$E_{RMSE}=\sqrt{\frac{1}{n}\sum_{i=1}^{n}(y_{i}-\overset{\wedge }{y_{i}}{)}^{2}}$$(15)$$E_{MAPE}=\frac{1}{n}\sum_{i=1}^{n}\frac{|y_{i}-\overset{\wedge }{y_{i}|}}{y_{i}}\times 100\%$$

The prediction performance of each model is shown in Table 3. From the data in Table 3, it is evident that the WOA-TCN-Attention model has the smallest prediction error, indicating that this model performs the best, followed by the TCN-Attention and TCN models, while the LSTM model has the worst prediction performance. Moreover, by comparing the computational time of various models, it was found that the proposed WOA-TCN-Attention model not only improves prediction accuracy but also ensures lower computational time consumption, resulting in better overall performance. This is because an attention mechanism was integrated into the TCN to learn the feature vectors of time-series data. By leveraging the attention mechanism, greater weight was assigned to key factors that significantly influence changes in gearbox oil sump temperature. Multiple variable inputs were utilized to predict the gearbox oil sump temperature. Additionally, the Whale Optimization Algorithm (WOA) was employed to optimize the hyperparameters of the TCN-Attention model, thereby enhancing its prediction accuracy.

This paper utilizes the statistical characteristics of residual values to set fault alarm and warning thresholds. Gearbox failure is predicted based on whether the absolute value of the residual exceeds the predefined thresholds. When the residual follows a normal distribution, the probability of the residual value falling outside [u−3∗δ,u+3∗δ] the target value is very small. The values of the u thresholds and d alarm levels are determined based on the residuals between the actual and predicted values of the gearbox training data. When the target value falls within this range, it typically indicates that the gearbox’s operating status is abnormal. If a certain number of anomalies persist, especially those with a relatively low probability of occurrence and that fall outside the above range, they may be identified as outliers. Based on the principle of the normal distribution confidence interval, the fault alarm upper and lower threshold values are set as u±3δ, and the alarm’s upper and lower threshold values are u±2δ. The number of outliers can reflect the status of abnormal faults, thereby helping to determine whether the gearbox is experiencing instability. By calculating the residuals of the test sample data and the predicted data, the distribution characteristics of the residuals are obtained, and a normal distribution curve is fitted, as shown in Figure 10.

After calculating the parameters of the normal distribution, the data from the 24 h preceding the failure of the wind turbine are selected to predict the gearbox oil sump temperature. This allows the residual values of the gearbox oil sump temperature to be obtained, as shown in Figure 11. The dashed line in the figure represents the warning line, while the solid line represents the alarm line. After sample point 300, the gearbox oil sump temperature exceeds both the fault warning and alarm thresholds. However, the gearbox fails near sample point 1400, by which time the oil sump temperature has already exceeded the alarm limits. Considering that samples are taken every minute, it is possible to predict signs of gearbox failure 18 h and 33 min in advance. This method demonstrates the accuracy of the proposed early warning model and provides wind turbine operation and maintenance personnel with sufficient time for repairs, thereby reducing downtime and lowering maintenance costs.

### 4.2. Implementation of Intelligent Operation and Maintenance for Wind Turbines Based on Digital Twin and Multi-Source Data Fusion

In order to achieve intelligent operation and maintenance (O&M) and optimized management of wind turbines, high-precision sensors are installed at key locations on the wind turbine to collect real-time data on critical operational parameters. These parameters include the temperature at the front end and rear end of the gearbox high-speed shaft, gearbox inlet oil temperature, generator active power, gearbox cooling water temperature, gearbox inlet pressure, gearbox oil sump temperature, and more. With these sensors, the operating status of the wind turbine can be comprehensively monitored, ensuring that all indicators remain within a safe and efficient operating range. To ensure the efficient use of multi-source data, the collected data are transmitted in real time via efficient data transmission channels to the cloud platform. On the cloud platform, the data undergo integration, cleaning, and preprocessing, and advanced data fusion technologies are employed to integrate and display data from different sensors. This not only ensures data consistency and integrity but also provides a reliable foundation for subsequent analysis and decision making, as shown in Figure 12:

To construct the digital twin model of the wind turbine gearbox, a 3D project is created in the Unity 3D game engine. The 3D model created in NX12.0 is converted into the .fbx format, which is recognized by Unity 3D, using C4D, and then imported into the project’s Asset package. The assembly relationships between various components are established using their position coordinates. Since there are interlocking relationships between the mechanical parts of the gearbox, these interlocking relationships cause subcomponents to inherit changes in position, rotation, movement, and scaling from the parent components. Therefore, the hierarchical relationships of the components are defined in the Unity 3D Hierarchy, enabling the interactivity between components in the virtual scene. Figure 13 and Figure 14 show the assembly and hierarchical relationships of the gearbox and the complete digital twin model of the wind turbine gearbox.

In order to achieve intelligent monitoring of the wind turbine system and predict its future state, the time-series data of seven variables shown in Table 1 are input into the TCN-Attention model. The model calculates and outputs the predicted value of the gearbox oil temperature. The Whale Optimization Algorithm (WOA) is used to optimize the model’s hyperparameters, including the number of convolutional kernels, dilation factor, and learning rate, further improving the model’s prediction accuracy. The optimization aims to minimize the fitness value as the objective for hyperparameter adjustment. The gearbox fault warning process based on the WOA-optimized TCN-Attention model is shown in Figure 15.

The pre-trained WOA-TCN-Attention fault diagnosis algorithm model is integrated into the digital twin monitoring system of wind turbine gearboxes. This algorithm model combines the Whale Optimization Algorithm (WOA), Temporal Convolutional Network (TCN), and Attention Mechanism, effectively processing and analyzing the large amounts of multi-dimensional data collected during the operation of wind turbines. As a result, the digital twin system not only achieves precise real-time monitoring of the gearbox’s status but also possesses the ability to predict future gearbox oil temperature at specific time points.

Specifically, using historical data and real-time collected data, the WOA algorithm optimizes the model’s parameters, while the TCN captures the temporal relationships within the oil temperature data. The attention mechanism then automatically adjusts the importance of key features, improving both prediction accuracy and model stability. The system can accurately predict the trend of gearbox oil temperature changes within a certain time frame, providing powerful decision support for maintenance personnel.

When the system detects abnormal fluctuations in key parameters such as oil temperature or predicts potential equipment failures, the algorithm immediately triggers an early warning mechanism, notifying the management team in real time and assisting maintenance personnel in taking preventive measures before issues occur. This helps avoid significant operational disruptions caused by equipment failures. Additionally, the system generates detailed fault analysis reports based on real-time data and prediction results, assisting maintenance personnel in pinpointing issues more precisely and developing corresponding maintenance and servicing plans.

This method significantly enhances the intelligent operation and maintenance (O&M) capabilities of wind turbines. Through early warning and optimized scheduling, wind turbines can minimize unplanned downtime, optimize the allocation of O&M resources, and improve the operational efficiency and reliability of the turbines.

Relying on Visual Studio 2019, PyCharm Community Edition 2023, and Unity 3D for the UI, the collected real-time data are input into the trained LSTM network model to compute the predicted values. The residuals are then calculated by comparing the predicted and real values. When the residuals exceed a threshold, the digital twin system’s backend script uses the GameObject.SetActive() function to activate the fault warning UI pop-up, displaying fault information to the staff and promptly alerting them to intervene and inspect. When a fault is detected, the human–machine interface will display a red exclamation mark and show the fault information on the main interface. As shown in Figure 16, the system displays the real-time changes in oil temperature, the predicted trends, and early warning information, further enhancing the work efficiency and decision-making accuracy of maintenance personnel.

## 5. Conclusions

This paper systematically explores an intelligent operation and maintenance (O&M) method based on digital twin and multi-source data fusion. A novel remote intelligent O&M framework for wind turbines based on digital twin technology is proposed, enriching the methodology of wind turbine condition monitoring and O&M. An innovative WOA-TCN-Attention algorithm model is introduced, which incorporates an attention mechanism into the Temporal Convolutional Network (TCN) to learn the feature vectors of time series. The attention mechanism assigns greater weights to key influencing factors, thereby enhancing the computational performance of the model. Simultaneously, the optimal hyperparameters of the model network are intelligently selected based on the Whale Optimization Algorithm (WOA), further improving the model’s performance. Experimental validation demonstrates that the proposed WOA-TCN-Attention model exhibits the smallest prediction error and significantly outperforms other compared models in terms of performance. Finally, a digital twin system for intelligent O&M of wind turbines is developed, which deeply integrates the proposed computational model with the digital twin model to achieve early warning of gearbox faults. The application of the method proposed in this study can significantly improve the transparency, efficiency, and accuracy of wind turbine O&M, providing valuable repair time for O&M personnel and effectively avoiding economic losses caused by wind turbine downtime. This research offers an efficient and feasible reference solution for the remote intelligent O&M of wind turbines.

Although the proposed method in this study has improved the intelligent operation and maintenance (O&M) level of wind turbines to some extent, there are still areas that require further refinement. While the study enables early fault warnings to guide O&M personnel in addressing potential issues in advance, it does not yet support the intelligent generation of comprehensive maintenance recommendations and plans. Therefore, future research should focus on developing more advanced algorithm models to enhance prediction accuracy and robustness. Additionally, it is essential to establish a real-time correlation function between the model’s predictions and the degradation trends of wind turbines. Based on these trends, automated maintenance plans and decision-making suggestions can be generated, further improving the intelligent O&M capabilities of wind turbines.

## Figures and Tables

**Figure 1 sensors-25-01972-f001:**
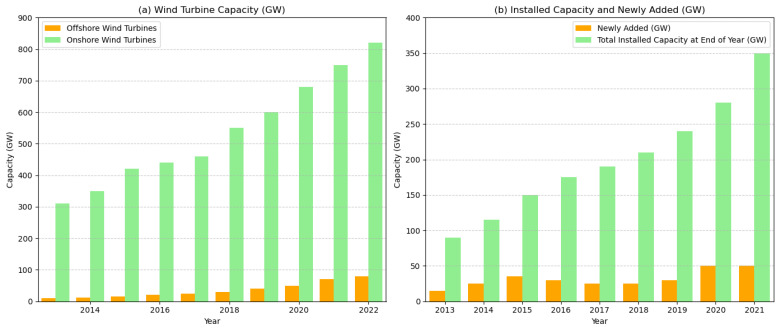
(**a**) Global wind turbine installed capacity. (**b**) China’s wind turbine installed capacity.

**Figure 2 sensors-25-01972-f002:**
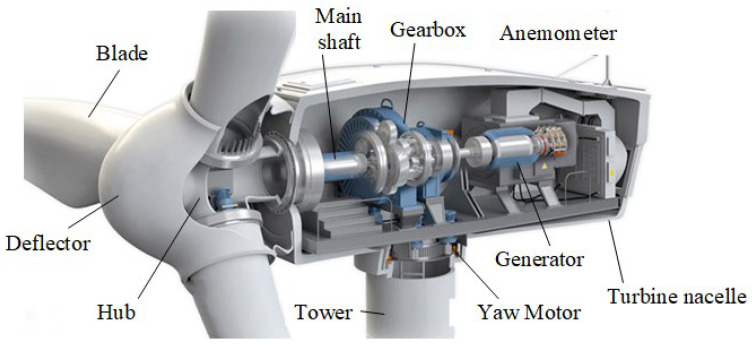
The basic structure of a wind turbine.

**Figure 3 sensors-25-01972-f003:**
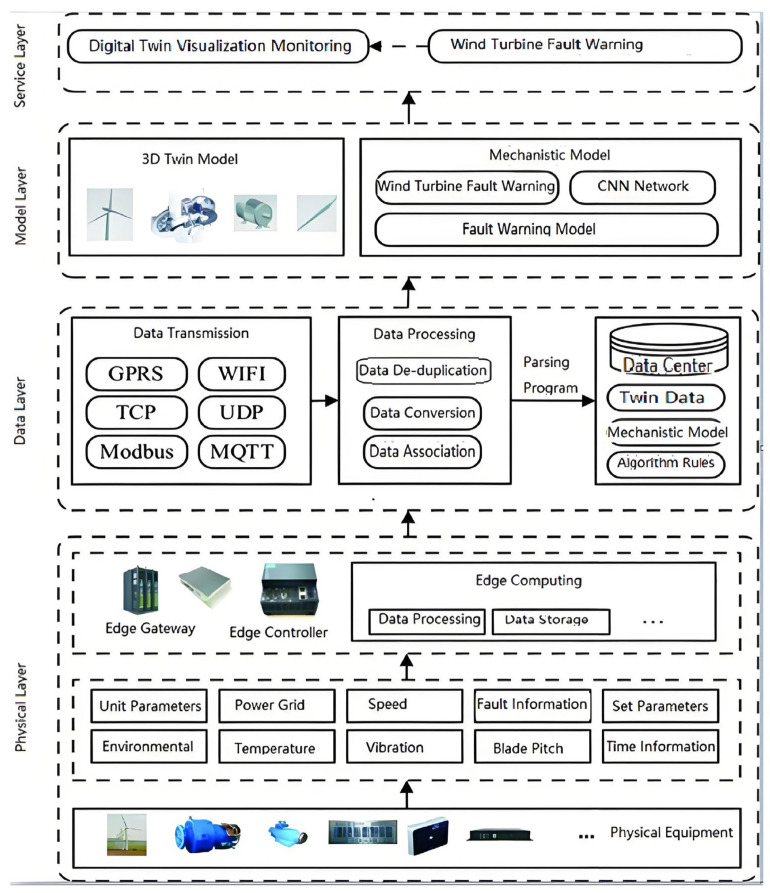
This intelligent operation and maintenance digital twin system for wind turbines.

**Figure 4 sensors-25-01972-f004:**
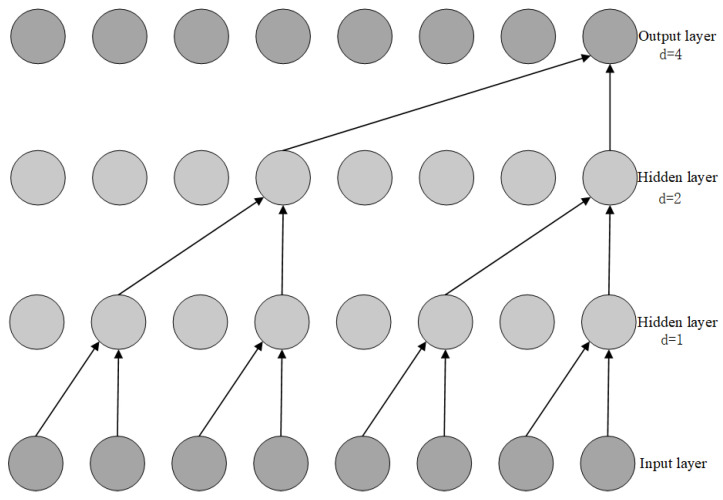
The structure of the causal dilated convolution network in TCN.

**Figure 5 sensors-25-01972-f005:**
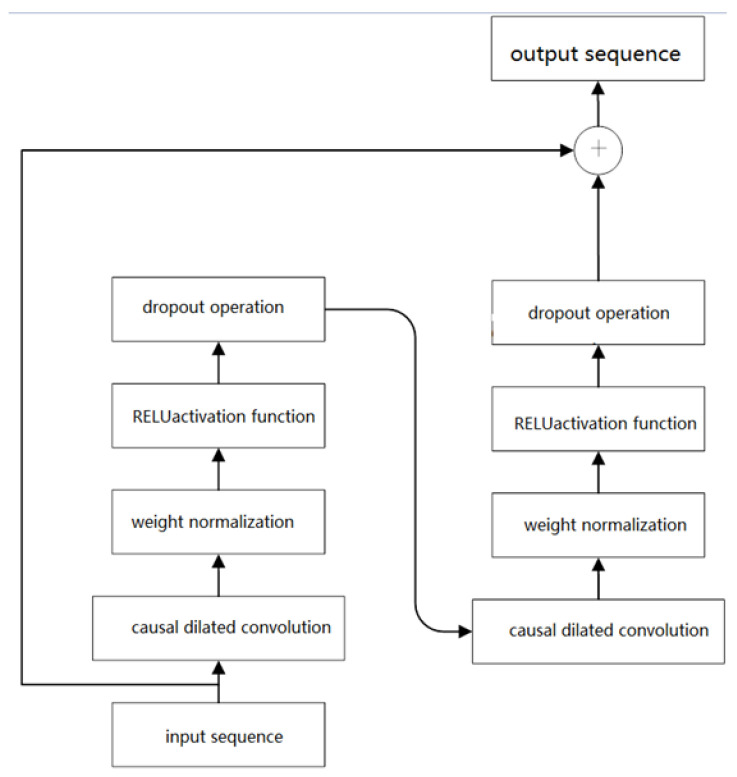
Residual connection module in the TCN network.

**Figure 6 sensors-25-01972-f006:**
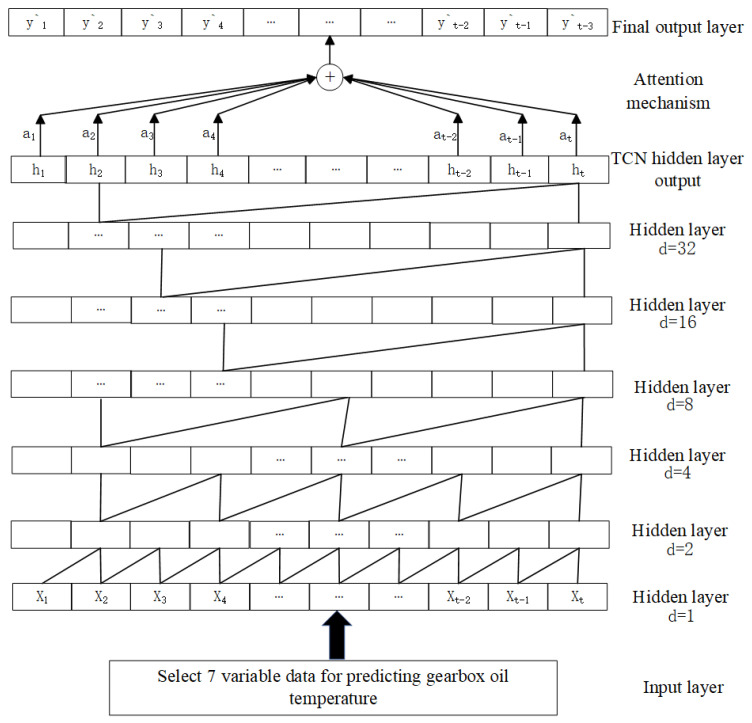
TCN-Attention structure of the model.

**Figure 7 sensors-25-01972-f007:**
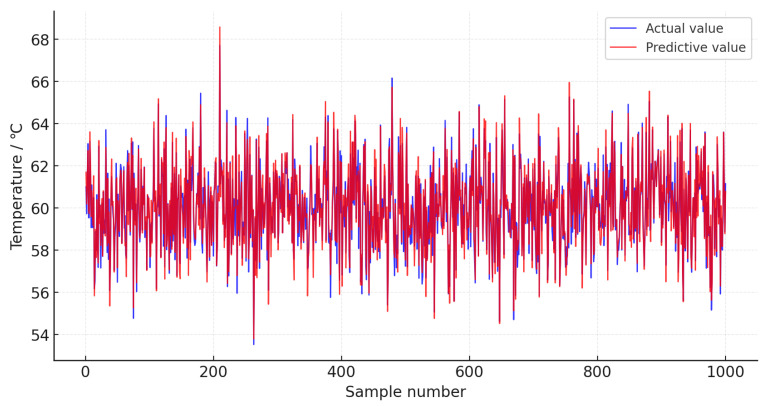
WOA-TCN-Attention Model gearbox oil tank temperature prediction and actual results.

**Figure 8 sensors-25-01972-f008:**
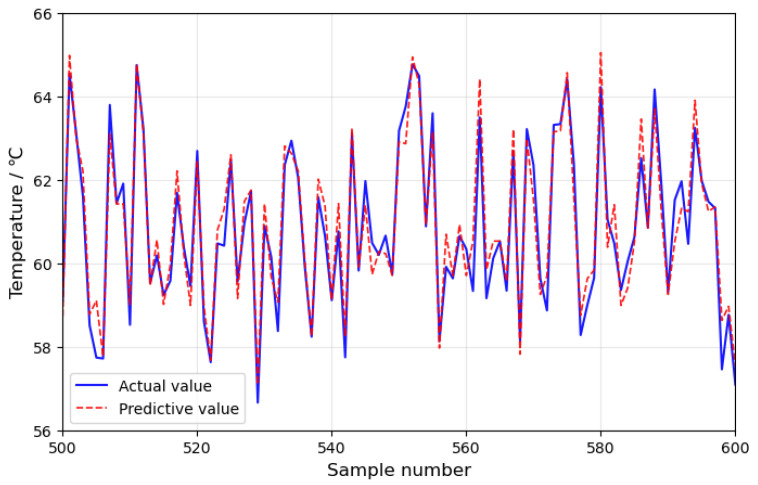
Partial data sample predictions and actual results.

**Figure 9 sensors-25-01972-f009:**
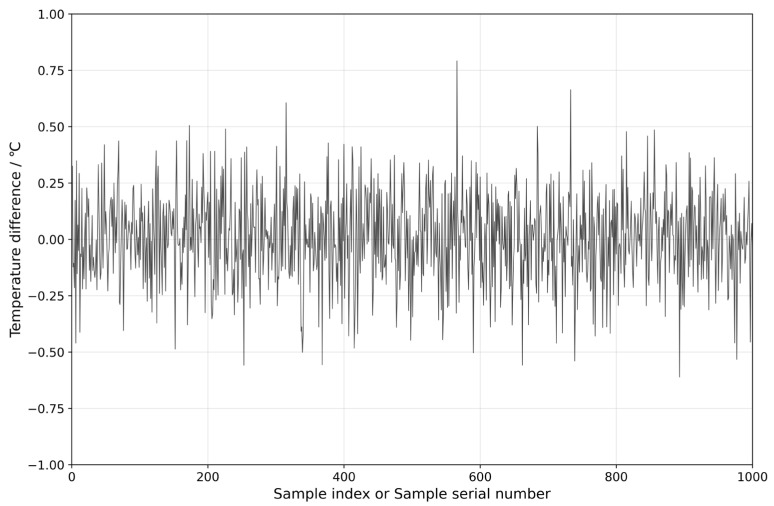
The residuals between the predicted values and the actual values.

**Figure 10 sensors-25-01972-f010:**
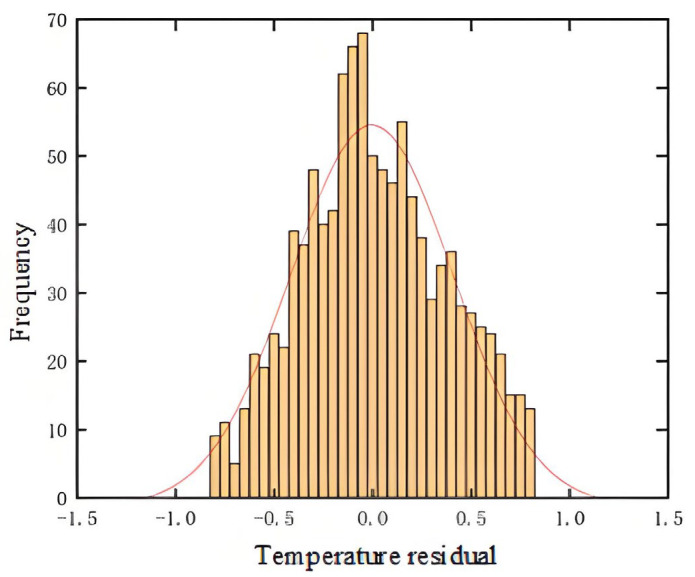
Residual Value Normal Distribution Fitting Histogram.

**Figure 11 sensors-25-01972-f011:**
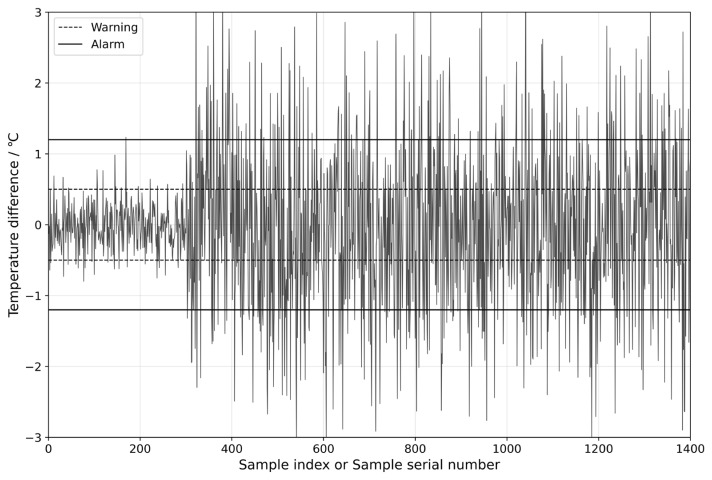
Fault Warning Threshold Interval.

**Figure 12 sensors-25-01972-f012:**
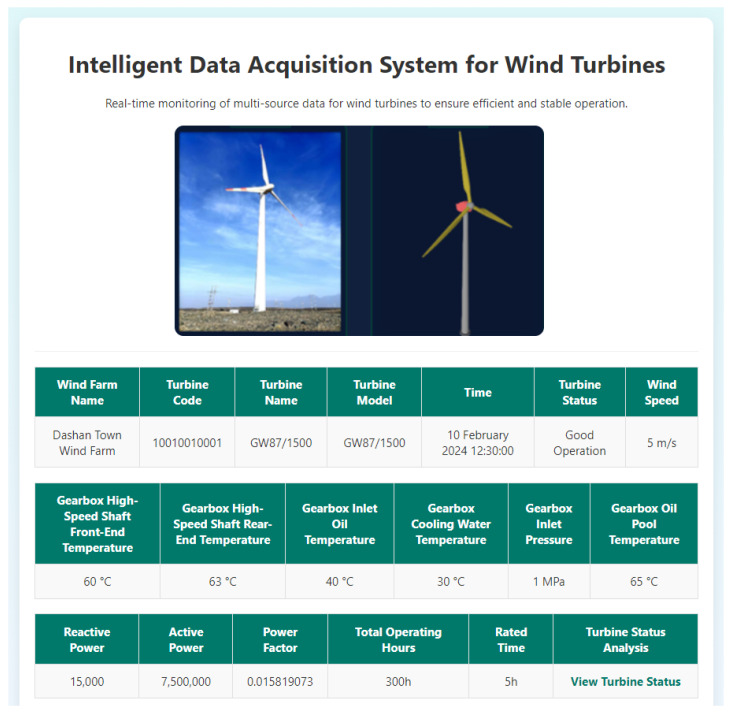
Real-time Data Collection and Monitoring of Multi-source Data for Wind Turbine Systems.

**Figure 13 sensors-25-01972-f013:**
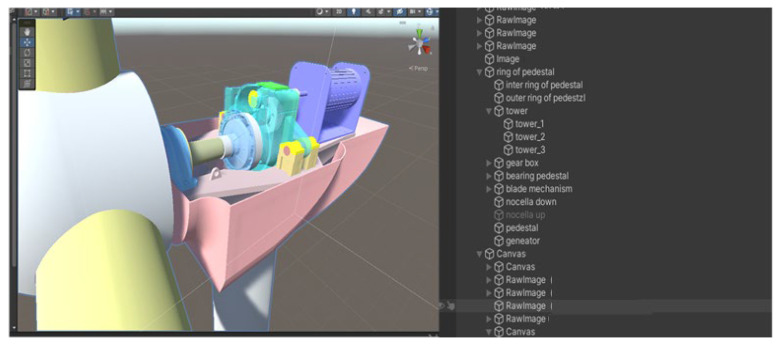
Hierarchical Configuration of the Digital Twin Model of the Wind Turbine Gearbox.

**Figure 14 sensors-25-01972-f014:**
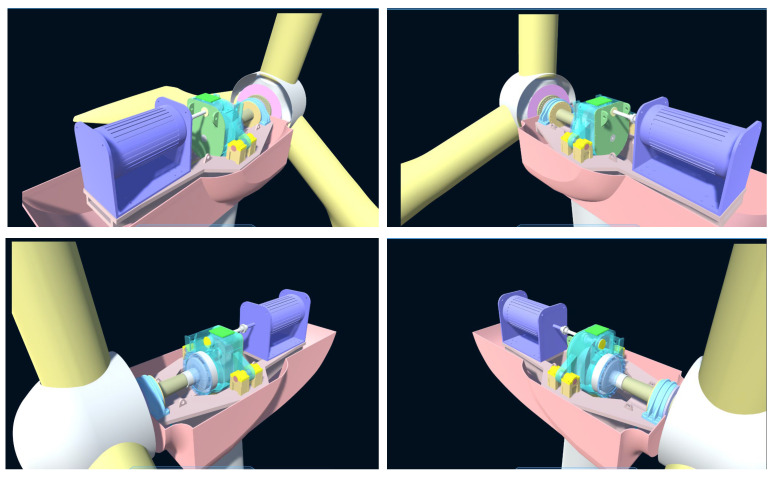
Digital Twin Model of the Wind Turbine Gearbox from Different Perspectives.

**Figure 15 sensors-25-01972-f015:**
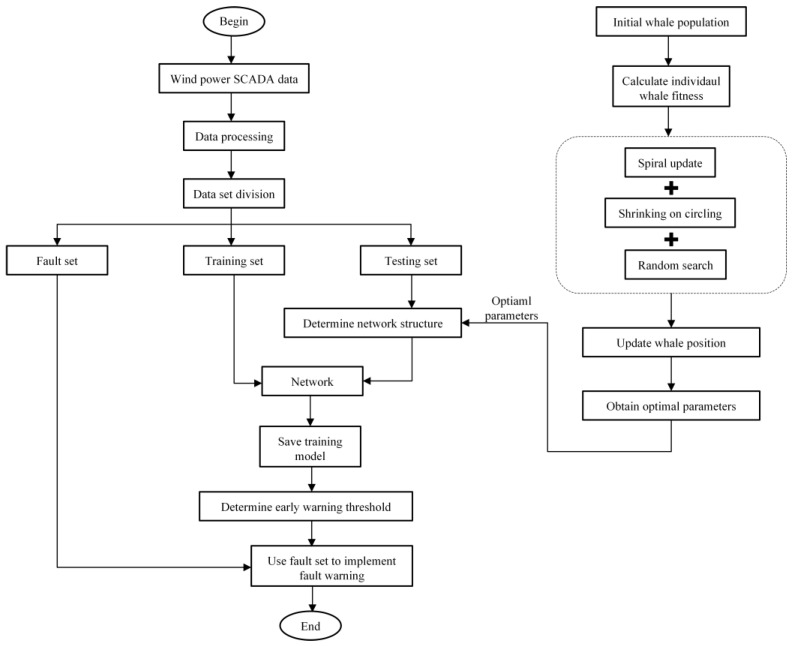
Early warning process.

**Figure 16 sensors-25-01972-f016:**
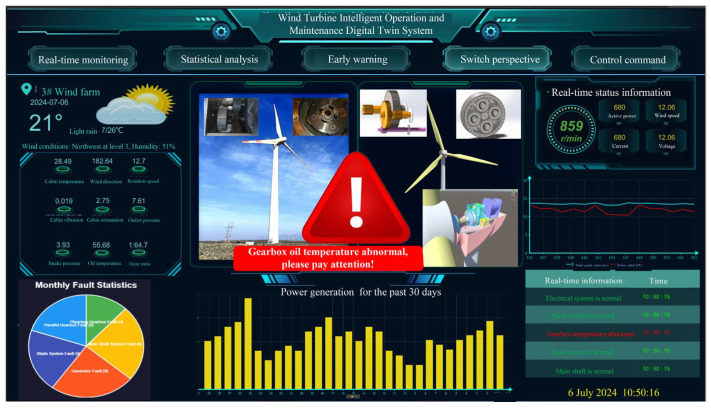
Intelligent operation and maintenance digital twin system for wind turbine gearbox.

**Table 1 sensors-25-01972-t001:** SCADA Variables for Training the Model.

Index	SCADA Variables	MIC Correlation Coefficient
A0	Front-end temperature of the gearbox high-speed shaft	0.7855
A1	Rear-end temperature of the gearbox high-speed shaft	0.8380
A2	Gearbox inlet oil temperature	0.7538
A3	Active power of the generator	0.6359
A4	Gearbox cooling water temperature	0.7221
A5	Gearbox inlet pressure	0.6329
A6	Gearbox oil sump temperature	1.0000

**Table 2 sensors-25-01972-t002:** Optimal hyperparameter combination.

Name	Learning Rate	Filters	Convolution Kernels	Dilation List
Value	0.006375	63	2	[1,2,4,8,16,32]

**Table 3 sensors-25-01972-t003:** Comparison of prediction results of 4 models.

Prediction Model	EMAE	ERMSE	EMAPE	Calculation Time
LSTM	0.3940	0.5728	0.6546	500 ms
TCN	0.2881	0.4554	0.4969	100 ms
TCN-Attention	0.2357	0.3968	0.3872	200 ms
WOA-TCN-Attention	0.2188	0.3685	0.3616	200 ms

## Data Availability

The data presented in this study are available upon request from the corresponding author.

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
