# Peer review of "Intelligent Operation and Maintenance of Wind Turbines Gearboxes via Digital Twin and Multi-Source Data Fusion"

_sensors, 2025, doi:10.3390/s25071972_

Round 1
Reviewer 1 Report
Comments and Suggestions for Authors
The authors should present the parameters of the gearbox used in this article.
The authors should present the different failures in a wind turbine chain with the percentage of the failure of each component. This information is retrieved via what type of sensor?
How many sensors were used and how much data was analyzed and sampled?
In Table 3, the authors give “EMAE, ERMSE and EMAPE”, they have to give the calculation time for each method.
What is the accuracy of such a method and what is the calculation time?
The quality of some figures needs to be improved.
References should be enriched. Authors can use the two articles below to improve their article.
1) Wind speed forecasting using machine learning approach based on meteorological data-a case study, Energy and Environment Research 2022
2) Wind Turbine Gearbox Failure Identification With Deep Neural Networks, IEEE Transactions on Industrial Informatics, 2017
Comments on the Quality of English LanguageSome modifications are needed.
Author Response
Reviewer 1
Thank you for your thorough review and valuable suggestions on our manuscript. Your insightful comments have significantly improved the quality of our research and deepened our understanding of the topic. We greatly appreciate the time and effort you dedicated to our work, and we have made revisions accordingly based on your suggestions.
The following are the relevant details of manuscript revisions. For your convenience,We uploaded both the marked version of the manuscript with revisions and the final clean version. The changes in the marked version have been highlighted in red and yellow.
Comments
Comments and Suggestions for Authors
Comment 1.The authors should present the parameters of the gearbox used in this article.
Response1:
The description of the gearbox parameters has been added in Section 4.1.
Comment 2.The authors should present the different failures in a wind turbine chain with the percentage of the failure of each component. This information is retrieved via what type of sensor?How many sensors were used and how much data was analyzed and sampled?
Response2:
The explanation of the relevant information has been added in the Section 4.1.
Comment 3.In Table 3, the authors give “EMAE, ERMSE and EMAPE”, they have to give the calculation time for each method.What is the accuracy of such a method and what is the calculation time?
Response3:
This is a regression prediction, and the model performance is expressed and evaluated using parameters such as variance (EMAE, ERMSE, and EMAPE). Additionally, a comparison of the computational time for each method has been included, as shown in Table 3.
Comment 4.The quality of some figures needs to be improved.
Response4:
All the figures have been carefully checked, and their clarity and quality have been improved.
Comment 5.References should be enriched. Authors can use the two articles below to improve their article.
1) Wind speed forecasting using machine learning approach based on meteorological data-a case study, Energy and Environment Research 2022
2) Wind Turbine Gearbox Failure Identification With Deep Neural Networks, IEEE Transactions on Industrial Informatics, 2017
Response4:
Thank you for the provided references, which have been cited in the text.
Reviewer 2 Report
Comments and Suggestions for Authors
This paper presents an Intelligent Operation and Maintenance Method for Wind
Turbines Based on Digital Twin Technology and Multi-Source 3 Data Fusion. This article needs to be revised before it can be considered for acceptance. Machine learning methods related to pattern recognition should be supplemented in the introduction part, including 10.1016/j.inffus.2024.102554, 10.1016/j.eswa.2024.125214 and so on.The image in the text is unclear, please adjust it clearly. Further analysis should be conducted to show the rationality of model parameter selection. How robust and computationally efficient are the proposed methods?
N/A
Author Response
Reviewer 2
Comments
Comments and Suggestions for Authors
This paper presents an Intelligent Operation and Maintenance Method for Wind
Turbines Based on Digital Twin Technology and Multi-Source 3 Data Fusion. This article needs to be revised before it can be considered for acceptance. Machine learning methods related to pattern recognition should be supplemented in the introduction part, including 10.1016/j.inffus.2024.102554, 10.1016/j.eswa.2024.125214 and so on.The image in the text is unclear, please adjust it clearly. Further analysis should be conducted to show the rationality of model parameter selection. How robust and computationally efficient are the proposed methods?
Response:
Thank you very much for the valuable suggestions. I have made the following modifications:
- I have added the recommended references as well as some additional ones in the introduction.
- All the figures in the paper have been reviewed, and clearer versions have been provided.
- The method principle and objective function for parameter optimization have been explained at the location of Table 2. At Table 3, an analysis and comparison of model performance have been added (minimizing error while maintaining lower computational time), highlighting the advantages of the method used in this paper.
Reviewer 3 Report
Comments and Suggestions for Authors
Overview
This manuscript proposes a wind turbine intelligent operation and maintenance method based on digital twin technology and multi-source data fusion. The approach aims to improve the operational efficiency and predictive maintenance capabilities of wind turbines by developing a digital twin system. However, there are several aspects that can be improved.
First, the literature review section needs to discuss key studies more comprehensively, including digital twin-based framework for wind turbine structures, and model structure, data processing methods, and their strengths and weaknesses, so that readers can better understand the substantive progress of relevant studies. In addition, it should be briefly stated in which aspects the method of this paper has been improved or innovated, so as to highlight the uniqueness and value of the work in this paper.
Second, the clarity of target selection and methodology evaluation needs to be improved. The paper should establish a reliable evaluation method to evaluate the performance of the proposed method, especially in terms of data integration performance and prediction accuracy.
Third, the structure of the text and the fluency of the argument need to be improved to improve the reading experience and comprehension. It is recommended that the author provide more supporting evidence and contrast different viewpoints to provide a broader and more complete perspective to support the claims of the article.
Overall, this paper has made some progress in improving the intelligent operation and maintenance level of wind turbines, but there is still room for improvement.
In-depth Comments
Major (Critical)
- [Whole article] Please improve English academic writing as the statement and argument are being devalued since they are presented colloquially and with shallow summary.
- [Whole article] There are many arguments lack of supportive evidence, please prove the claims before concluding them.
- [Section 2] It is suggested to improve the content writing of related work, which mainly has the following problems.
- (1) Comprehensiveness of literature review
- For some key studies, the model structure, data processing methods and advantages and disadvantages can be discussed in detail, rather than simply listed, so that readers can better understand the substantive progress of relevant research
- (2) Rationality of document classification
- The current literature review can be further refined, and it is suggested to discuss the recent relevant studies of researchers in separate sections. For example, for the application of digital twin technology, a separate section could be created to discuss the unique advantages and challenges of digital twin technology in wind turbine operation and maintenance.
- (3) Critical analysis of existing research
- While pointing out the shortcomings of existing research, it is possible to analyze the causes of these shortcomings in more depth, as well as possible solutions. For example, for gradient problems and long training times of deep learning models, some of the latest optimization algorithms or model architecture improvements can be discussed, as well as their potential applications in wind turbine monitoring.
- (4) Relevance to the work in this paper
- In the course of the review, the relationship between the work in this paper and existing research could be mentioned more frequently. For example, when discussing each existing method, you can briefly state where the method in this paper has been improved or innovated to highlight the uniqueness and value of the work in this paper. For some researches similar to the method presented in this paper, the differences between them and the method presented in this paper in algorithm design, data processing and model optimization can be analyzed in detail, so as to further highlight the advantages of the method presented in this paper.
Minor (Improvement)
- [Section 1, line 36] It is recommended to consider the latest data from the 2024 GLobal Wind Report and cite it in the references.
- [Section 1, line 63-65] The existing problems of the method lack of reference support, it is suggested to supplement.
- [Section 1, line 66-67] “New monitoring and fault prediction methods”lacks reference support, it is suggested to supplement
- [Section 1, line70-71] The problem of high delay in fault prediction lacks reference support, so it is recommended to supplement.
- [Section 2, line 184-192]These limitations of existing studies were not highlighted in the process of literature review, so it is difficult to support these views. It is suggested to sort out the existing research work in a targeted way.
- [Section 2, line 187] Please explain why the temperature variation of the gearbox oil pool is the target variable of this study.
- [Section 2, line 204-205] The content of case verification need not be mentioned here and can be deleted.
- [Section 3.1, line 378-381] Please carefully check that mathematical symbols (such as summation symbols) and formulas are formatted in accordance with the specification.
- [Section 3.2, line 398-402] Please carefully check that mathematical symbols and formulas are formatted in accordance with the specification.
- [Section3.3, line411] The section number is wrong, please check it, it should be changed to 3.3.3.
- [Section3.3, line 420] Please explain whether the “Attention layer”and “Attention mechanism” are the same.
- [Section 3.3.3, line 421-422]It is recommended to explain above that the output layer is a fully connected layer for readers to understand.
- [Section 3.4, line 458] Please carefully check that mathematical symbols and formulas are formatted in accordance with the specification.
- [Section 3.3.4, line 462-463] Please carefully check that mathematical symbols and formulas are formatted in accordance with the specification.
- [Section 1, line 485] Please carefully check that mathematical symbols and formulas are formatted in accordance with the specification.
- [Section 1, line 529-531] Please carefully check that mathematical symbols (such as summation symbols) and formulas are formatted in accordance with the specification.
- [Section 1, line 547-548]The difference between the two thresholds is not specified.Please refer to the preceding alarm threshold and warning threshold and explain the difference.
- [Section 2, line 619-620]Please provide a more detailed description of the predictive capabilities of the system to avoid empty statements.
- [Section 2, line 625-626]Please provide a more detailed explanation. The cases used in the study do not well reflect the prediction ability of the system. It is possible to adjust the selection of cases or add more cases to reflect the prediction performance of the system to avoid empty statements.
- [Section 2, line 631-634]The existing cases cannot prove that the system can generate specific fault analysis reports. Please clarify the data that the system can provide to maintenance personnel and the contents of fault analysis reports based on the cases to avoid empty statements.
- [Section1 , Figure 1] It is recommended to improve the figure clarity, unify the size of the picture frame, and pay attention to the font size of the legend.
- [Section3, Figure 2] It is recommended to improve the figure clarity and uniform font size.
- [Section3, Figure 3] Please improve the clarity of the figure, unify the font size, and check whether the chart format meets the specification. Whether the mechanism model of the model layer in the figure contains three parts that are juxtaposed, please elaborate.
- [Section3, Figure 4] It is recommended to improve the figure clarity and explain the Input layer,Hidden layer, and Output layer.
- [Section3, Figure 5] Please improve the clarity of the figure, standardize the format and font of the flowchart, and explain the specific details of the residual network structure as necessary.
- [Section3, Figure 6] Please improve the clarity of the
- [Section4, Figure 7] Please improve the clarity of the And the two curves in the figure are recommended to choose different lines to distinguish, to avoid confusion when the article is printed in black and white.
- [Section4, Figure 8] Please improve the sharpness of the figure and select a suitable line to distinguish the two curves.
- [Section4, Figure 9] Please improve the sharpness of the
- [Section4, Figure 10] Please improve the clarity and adjust the font of the horizontal and vertical headings.
- [Section4, Figure 11] Please improve the clarity and add necessary warning line and alarm line legends.
English writing can be improved by seeking professional editing services or language tools.
Author Response
Reviewer 3
Comments
Comments and Suggestions for Authors
Overview
This manuscript proposes a wind turbine intelligent operation and maintenance method based on digital twin technology and multi-source data fusion. The approach aims to improve the operational efficiency and predictive maintenance capabilities of wind turbines by developing a digital twin system. However, there are several aspects that can be improved.
First, the literature review section needs to discuss key studies more comprehensively, including digital twin-based framework for wind turbine structures, and model structure, data processing methods, and their strengths and weaknesses, so that readers can better understand the substantive progress of relevant studies. In addition, it should be briefly stated in which aspects the method of this paper has been improved or innovated, so as to highlight the uniqueness and value of the work in this paper.
Second, the clarity of target selection and methodology evaluation needs to be improved. The paper should establish a reliable evaluation method to evaluate the performance of the proposed method, especially in terms of data integration performance and prediction accuracy.
Third, the structure of the text and the fluency of the argument need to be improved to improve the reading experience and comprehension. It is recommended that the author provide more supporting evidence and contrast different viewpoints to provide a broader and more complete perspective to support the claims of the article.
Overall, this paper has made some progress in improving the intelligent operation and maintenance level of wind turbines, but there is still room for improvement.
Response:
Based on your suggestions, we have added some new references in the literature review section to help readers better understand the substantial progress in related research. Additionally, we included a comparison and analysis of existing methods and the method proposed in this paper, emphasizing the uniqueness and value of our work. In terms of model evaluation, as shown in Table 3, we have presented and explained the comparison of error and computational time for various methods, highlighting the superior performance of the model proposed in this paper.
In-depth Comments
Major (Critical)
- [Whole article] Please improve English academic writing as the statement and argument are being devalued since they are presented colloquially and with shallow summary.
- [Whole article] There are many arguments lack of supportive evidence, please prove the claims before concluding them.
- [Section 2] It is suggested to improve the content writing of related work, which mainly has the following problems.
- (1) Comprehensiveness of literature review
- For some key studies, the model structure, data processing methods and advantages and disadvantages can be discussed in detail, rather than simply listed, so that readers can better understand the substantive progress of relevant research
- (2) Rationality of document classification
- The current literature review can be further refined, and it is suggested to discuss the recent relevant studies of researchers in separate sections. For example, for the application of digital twin technology, a separate section could be created to discuss the unique advantages and challenges of digital twin technology in wind turbine operation and maintenance.
- (3) Critical analysis of existing research
- While pointing out the shortcomings of existing research, it is possible to analyze the causes of these shortcomings in more depth, as well as possible solutions. For example, for gradient problems and long training times of deep learning models, some of the latest optimization algorithms or model architecture improvements can be discussed, as well as their potential applications in wind turbine monitoring.
- (4) Relevance to the work in this paper
- In the course of the review, the relationship between the work in this paper and existing research could be mentioned more frequently. For example, when discussing each existing method, you can briefly state where the method in this paper has been improved or innovated to highlight the uniqueness and value of the work in this paper. For some researches similar to the method presented in this paper, the differences between them and the method presented in this paper in algorithm design, data processing and model optimization can be analyzed in detail, so as to further highlight the advantages of the method presented in this paper.
Response:
In Sections 1 and 2, relevant key references have been added and analyzed. We have also included a comparison and analysis of the limitations of existing research and the advantages of this study. By analyzing and comparing the shortcomings of current research in Sections 1 and 2, the problems this study aims to address and the advantages of the proposed method have been introduced.
Minor (Improvement)
- [Section 1, line 36] It is recommended to consider the latest data from the 2024 GLobal Wind Report and cite it in the references.
- [Section 1, line 63-65] The existing problems of the method lack of reference support, it is suggested to supplement.
- [Section 1, line 66-67] “New monitoring and fault prediction methods”lacks reference support, it is suggested to supplement
- [Section 1, line70-71] The problem of high delay in fault prediction lacks reference support, so it is recommended to supplement.
- [Section 2, line 184-192]These limitations of existing studies were not highlighted in the process of literature review, so it is difficult to support these views. It is suggested to sort out the existing research work in a targeted way.
- [Section 2, line 187] Please explain why the temperature variation of the gearbox oil pool is the target variable of this study.
- [Section 2, line 204-205] The content of case verification need not be mentioned here and can be deleted.
- [Section 3.1, line 378-381] Please carefully check that mathematical symbols (such as summation symbols) and formulas are formatted in accordance with the specification.
- [Section 3.2, line 398-402] Please carefully check that mathematical symbols and formulas are formatted in accordance with the specification.
- [Section3.3, line411] The section number is wrong, please check it, it should be changed to 3.3.3.
- [Section3.3, line 420] Please explain whether the “Attention layer”and “Attention mechanism” are the same.
- [Section 3.3.3, line 421-422]It is recommended to explain above that the output layer is a fully connected layer for readers to understand.
- [Section 3.4, line 458] Please carefully check that mathematical symbols and formulas are formatted in accordance with the specification.
- [Section 3.3.4, line 462-463] Please carefully check that mathematical symbols and formulas are formatted in accordance with the specification.
- [Section 1, line 485] Please carefully check that mathematical symbols and formulas are formatted in accordance with the specification.
- [Section 1, line 529-531] Please carefully check that mathematical symbols (such as summation symbols) and formulas are formatted in accordance with the specification.
- [Section 1, line 547-548]The difference between the two thresholds is not specified.Please refer to the preceding alarm threshold and warning threshold and explain the difference.
- [Section 2, line 619-620]Please provide a more detailed description of the predictive capabilities of the system to avoid empty statements.
- [Section 2, line 625-626]Please provide a more detailed explanation. The cases used in the study do not well reflect the prediction ability of the system. It is possible to adjust the selection of cases or add more cases to reflect the prediction performance of the system to avoid empty statements.
- [Section 2, line 631-634]The existing cases cannot prove that the system can generate specific fault analysis reports. Please clarify the data that the system can provide to maintenance personnel and the contents of fault analysis reports based on the cases to avoid empty statements.
- [Section1 , Figure 1] It is recommended to improve the figure clarity, unify the size of the picture frame, and pay attention to the font size of the legend.
- [Section3, Figure 2] It is recommended to improve the figure clarity and uniform font size.
- [Section3, Figure 3] Please improve the clarity of the figure, unify the font size, and check whether the chart format meets the specification. Whether the mechanism model of the model layer in the figure contains three parts that are juxtaposed, please elaborate.
- [Section3, Figure 4] It is recommended to improve the figure clarity and explain the Input layer,Hidden layer, and Output layer.
- [Section3, Figure 5] Please improve the clarity of the figure, standardize the format and font of the flowchart, and explain the specific details of the residual network structure as necessary.
- [Section3, Figure 6] Please improve the clarity of the
- [Section4, Figure 7] Please improve the clarity of the And the two curves in the figure are recommended to choose different lines to distinguish, to avoid confusion when the article is printed in black and white.
- [Section4, Figure 8] Please improve the sharpness of the figure and select a suitable line to distinguish the two curves.
- [Section4, Figure 9] Please improve the sharpness of the
- [Section4, Figure 10] Please improve the clarity and adjust the font of the horizontal and vertical headings.
- [Section4, Figure 11] Please improve the clarity and add necessary warning line and alarm line legends.
Comments on the Quality of English Language
Response:
The required revisions and optimizations have been made as follows:
- Additional references have been added to enrich the discussion and support relevant points.
- Since the available data only includes the 2023 World Energy Statistics Yearbook data, the data in Figure 1 has not been updated. Additionally, relevant data from the GWEC 2024 Global Wind Report has been added in Section 1.
- All figures in the paper have been provided in higher quality and clearer versions.
- Other relevant items have been carefully reviewed and revised item by item.
Reviewer 4 Report
Comments and Suggestions for Authors
Dear Authors,
A detailed review of your article was conducted, identifying various opportunities for improvement, which are attached in a separate document. Below is a summary of the most relevant aspects that require attention to strengthen the quality and clarity of the study.
Although the article mentions O&M for wind turbines, the analysis focuses exclusively on the gearbox. This may be misleading for readers. It is recommended to modify the title accurately reflect the study's scope. Alternatively, to maximize its impact, the study could be expanded to rotating machinery in general, making it more relevant for various industrial applications.
The contribution of the study to scientific literature is not explicitly highlighted. It is essential to specify what advancements are achieved compared to previous research and what new knowledge is provided in the field of predictive maintenance for wind turbines.
There are unnecessary repetitions of concepts in several sections, which affect the document’s readability and clarity. It is recommended to condense the information and improve organization to facilitate comprehension.
Specialized terms are introduced without an adequate explanation or supporting references. The bibliography is insufficient to support the theoretical and methodological framework of the study. It is recommended to include citations from recent and relevant studies to strengthen the validity of the proposed model.
Several figures and equations are presented without a clear introduction or detailed interpretation. It is essential to explain how each equation relates to model optimization and what key information the figures provide.
Overall, the study addresses a relevant topic, but it needs a clearer focus, a stronger justification of its methodology, and a better-founded theoretical framework.
We hope these observations will contribute to improving the work and strengthening its impact on the scientific community.
Sincerely.

Dear Authors,
A detailed review of your article was conducted, identifying various opportunities for improvement, which are attached in a separate document. Below is a summary of the most relevant aspects that require attention to strengthen the quality and clarity of the study.
Although the article mentions O&M for wind turbines, the analysis focuses exclusively on the gearbox. This may be misleading for readers. It is recommended to modify the title accurately reflect the study's scope. Alternatively, to maximize its impact, the study could be expanded to rotating machinery in general, making it more relevant for various industrial applications.
The contribution of the study to scientific literature is not explicitly highlighted. It is essential to specify what advancements are achieved compared to previous research and what new knowledge is provided in the field of predictive maintenance for wind turbines.
There are unnecessary repetitions of concepts in several sections, which affect the document’s readability and clarity. It is recommended to condense the information and improve organization to facilitate comprehension.
Specialized terms are introduced without an adequate explanation or supporting references. The bibliography is insufficient to support the theoretical and methodological framework of the study. It is recommended to include citations from recent and relevant studies to strengthen the validity of the proposed model.
Several figures and equations are presented without a clear introduction or detailed interpretation. It is essential to explain how each equation relates to model optimization and what key information the figures provide.
Overall, the study addresses a relevant topic, but it needs a clearer focus, a stronger justification of its methodology, and a better-founded theoretical framework.
We hope these observations will contribute to improving the work and strengthening its impact on the scientific community.
Sincerely.
Author Response
Reviewer 4
Comments
Comments and Suggestions for Authors
A detailed review of your article was conducted, identifying various opportunities for improvement, which are attached in a separate document. Below is a summary of the most relevant aspects that require attention to strengthen the quality and clarity of the study.
Although the article mentions O&M for wind turbines, the analysis focuses exclusively on the gearbox. This may be misleading for readers. It is recommended to modify the title accurately reflect the study's scope. Alternatively, to maximize its impact, the study could be expanded to rotating machinery in general, making it more relevant for various industrial applications.
The contribution of the study to scientific literature is not explicitly highlighted. It is essential to specify what advancements are achieved compared to previous research and what new knowledge is provided in the field of predictive maintenance for wind turbines.
There are unnecessary repetitions of concepts in several sections, which affect the document’s readability and clarity. It is recommended to condense the information and improve organization to facilitate comprehension.
Specialized terms are introduced without an adequate explanation or supporting references. The bibliography is insufficient to support the theoretical and methodological framework of the study. It is recommended to include citations from recent and relevant studies to strengthen the validity of the proposed model.
Several figures and equations are presented without a clear introduction or detailed interpretation. It is essential to explain how each equation relates to model optimization and what key information the figures provide.
Overall, the study addresses a relevant topic, but it needs a clearer focus, a stronger justification of its methodology, and a better-founded theoretical framework.
We hope these observations will contribute to improving the work and strengthening its impact on the scientific community.
Sincerely.
Response:
- Based on your suggestion, the title has been modified to focus on the wind turbine gearbox.
- Additional references have been cited and analyzed in Sections 1 and 2, with a comparison between existing research and the method proposed in this paper, emphasizing the progress and advantages of this study.
- In Table 3, a detailed analysis and comparison of the results between the proposed method and other models have been provided, highlighting the advantages and advancement of our approach.
- Some viewpoints in the paper have been supported with references to strengthen their validity and authenticity.
- In Table 2, detailed explanations regarding the principles of parameter optimization, the objective function, and the selected parameters have been added.
Round 2
Reviewer 1 Report
Comments and Suggestions for Authors
The authors responded all my questions. But they have to correct the the reference number cited in their paper, they used the reference number 27 and 28 used but in the page 4 they wrote 26 and 27.
The authors can use the reference below to improve their paper:
-A smart multiphysics approach for wind turbines design in industry 5.0, ELSEVIER- Journal of Industrial Information Integration 2024
Some modifications and rewording are necessary.
Author Response
Thank you for your thorough review and valuable suggestions on our manuscript. Your insightful comments have significantly improved the quality of our research and deepened our understanding of the topic. We greatly appreciate the time and effort you dedicated to our work, and we have made revisions accordingly based on your suggestions. The following are the relevant details of manuscript revisions.Comments:
The authors responded all my questions. But they have to correct the the reference number cited in their paper, they used the reference number 27 and 28 used but in the page 4 they wrote 26 and 27.
The authors can use the reference below to improve their paper:
-A smart multiphysics approach for wind turbines design in industry 5.0, ELSEVIER- Journal of Industrial Information Integration 2024
Response:I have carefully reviewed all the references and cited the article you suggested.
Reviewer 2 Report
Comments and Suggestions for Authors
its ok
Author Response
Thank you for your thorough review and valuable suggestions on our manuscript. Your insightful comments have significantly improved the quality of our research and deepened our understanding of the topic. We greatly appreciate the time and effort you dedicated to our work.
Thank you!
I wish you a happy life as well!
Reviewer 4 Report
Comments and Suggestions for Authors
Dear Author,
In the first review, a document with observations was attached, which, for some reason, were not addressed. While some modifications have been made in response to other reviewers' comments, it is essential that the previously submitted observations are considered in this second review.
We appreciate your attention to this matter and remain available for any clarifications.
Best regards.

Author Response
Reviewer 4
Thank you for your thorough review and valuable suggestions on our manuscript. Your insightful comments have significantly improved the quality of our research and deepened our understanding of the topic. We greatly appreciate the time and effort you dedicated to our work, and we have made revisions accordingly based on your suggestions.
Comments and Response
The content in red font below is the response.
Title
The title of the article. " An intelligent Operation and Maintenance Method for Wind Turbines Basedon Digital Twin Technology and Multi-Source Data Fusion" can be improved with the followingobservations:
- -The title is too long, unnecessary words can be removed.
- -"Method" is not strictly necessary.
- -"Digital Twin Technology" can be reduced to "Digital Twin," as the term already implies a specifictechnology.
4.- The title does not clearly indicate whether the proposed method uses the digital twin and datafusion or if these are merely components of the solution.
Here's a possible suggestion:
"Intelligent Operation and Maintenance of Wind Turbines Gearboxes via Digital Twin and MultiSource Data Fusion"
Response:Thank you very much for your valuable suggestion. After careful consideration, we believe that the topic you proposed is more perfect, and we have decided to adopt it: "Intelligent Operation and Maintenance of Wind Turbines Gearboxes via Digital Twin and MultiSource Data Fusion."
Abstract
The abstract presents valuable information, but several aspects can be improved to enhance clarity,improve structure, and avoid redundancies. Here are some suggestions:
The abstract starts directly with the proposed solution but does not clearly state the specificproblem being addressed. lt is recommended to begin with a brief contextualization of theproblem.
The abstract mentions the development of a model based on WOA-TCN-Attention but doesnot clearly specify what data is used or how the model is evaluated. lt would be helpful toinclude a brief sentence indicating what type of operational turbine data is analyzed andwhat key metric is used to evaluate the model.
The last sentence, "The developed digital twin intelligent O&M system for wind turbinesenables real-time and efficient operation and maintenance." repeats information that hasalready been mentioned earlier.
Response:The abstract has been rewritten based on your suggestions, and the revised parts are highlighted in red in the manuscript.
1.-Introduction
The introduction begins with a general description of the global growth of wind energy, but thetransition to the maintenance problem is not sufficiently smooth. lt is recommended to include abridge sentence that connects the increase in installed capacity with the growing need for moreefficient maintenance strategies.
Although the introduction mentions that wind turbines face adverse conditions and that traditionalmaintenance methods are costly, it does not explicitly state the central problem that this studyaddresses. lt is recommended to include a guiding question or hypothesis so that the readerunderstands the article's focus from the beginning.
The introduction mentions traditional methods and new approaches (big data, neural networksmechanistic models) but does not explain why a new method is necessary. it is recommended toadd a brief justification for the approach adopted in the article.References are missing when describing the sizes of wind turbines and some technical data.
Response:1.Some of the revision suggestions have already been implemented in the previous revised version.
- A sentence has been added to serve as a transition between the installed capacity and the need for new maintenance strategy.
2.Related works
A detailed state-of-the-art review is provided on monitoring and fault prediction methods for windturbines, including approaches based on artificial intelligence, big data, and digital twin models.However, several aspects could be improved to enhance clarity, avoid redundancies, and improvestructure.
Too many individual studies are mentioned, describing their approaches without clearlyestablishing their differences, limitations, or how they relate to each other. lt isrecommended to synthesize the studies into categories instead of listing them individually.Although many solutions are described, there is no clear connection to the need for thisstudy. it is recommended to include a transition paragraph explaining the gaps in theliterature and how this work addresses them.
Some concepts and methodologies are repeated in different sections (e.g., Johansencointegration testing and the importance of intelligent turbine monitoring). lt isrecommended to eliminate redundancies and merge similar descriptions.Although the introduction mentions that the digital twin has been applied to wind turbines. itdoes not justify why this approach is superior to other traditional fault prediction models. Lt
is recommended to add an explicit comparison between the advantages of the digital twinand existing methods.
The last part provides excessive detail about the proposed model's architecture, which maybe more suitable forthe methodology section. ltis recommended to summarize the proposaland move technical details to the methodology.
Response:1.Some of the revision suggestions have already been implemented in the previous revised version.
- Redundant technical details in the last paragraph have been removed.
3.Materials and Methods
3.1. The Basic Structure of a Wind Turbine
The paragraph describing the wind turbine is clear and well-structured, but several aspects needimprovement:
Sizes and key components of wind turbines are mentioned without citing sources to supportthe information. References should be included for data on the turbine's structure,dimensions,and operation.
Response:References already cited
The text appears to refer to horizontal-axis wind turbines (HAWT) but does not explicitlyspecify this. lt should be clarified whether the description applies to all wind turbines or onlycertain models.
Response:Response: Yes, it is the HAWT type, which has been clearly stated in the manuscript.
The rotor is mentioned as including the nacelle cover, which is not entirely correct, as thenacelle is a separate structure.The description should be corrected, and terminology shouldbe more precise.
Response:The change has been made.
The conversion of mechanical energy into electrical energy is mentioned without explainingthe role of the generator and whether the transmission is direct or indirect.
It is stated that "the taller the tower, the greater the wind speed," but it is not explained that.the optimal height depends on factors such as terrain roughness and meteorologicalconditions.
Response:Relevant explanations have been added.
The last sentence about the gearbox is abrupt and does not connect well with the rest of theparagraph.
Response:The change has been made.
3.2. Remote Intelligent Operation and Maintenance Method for Wind Turbines Based on Digital TwinThe writing is somewhat repetitive, and some sentences could be reworded for greater clarityImproving fluency and precision is recommended.There is a lack of clarity in the relationship between concepts. The complexity of the wind turbineenvironment and the limitations of traditional methods are mentioned, but it is not specified whichaspects ofthese methods fail.
The "framework of the digital twin system for the energy internet" and the "five-dimensional digitaltwin model theory" are mentioned, but they are neither explained nor cited. Some phrases orinformation are repeated from previous sections: instead, the relationshin with the proposedmethodology should be addressed here.
It is stated that the digital twin system "fuses and analyzes multidimensional data." but it does notspecify what types of data are integrated or how they relate to fault detection. including examples offused data would be beneficial.
Many technologies are mentioned (e.g., Modbus, MQTT, WebGL, SQL Server, HHT, CNN) withoutbriefly explaining their purpose in the system. A short clarification should be added to preventreaders without specialized knowledge from getting lost.
Although each layer is described separately, it is not explained how they interact. A transitionsentence should be added at the end of each section to clarify the relationship between layers.
Mathematical models and algorithms are mentioned without references supporting their use in thecontext of wind turbines. Citations or references should be included to justify the choice of thesemodels.
The importance of cloud computing should be justified.
Response:
- The limitations of traditional methods have been described in detail in the introduction of the previous revised version.
- A reference (No. 54) has been added for the "energy internet digital twin system framework and the digital twin five-dimensional model theory".
- The term "multi-source data fusion" mentioned here is a macro-level method and framework. Specific data used are mentioned and explained in Sections 4.1 and 3.3.3.
- The purposes of many technologies mentioned in the article are explained within the text. For example, Modbus and MQTT are communication protocols used for data acquisition and transmission. SQL Server is a database used for data storage.
- As shown in the architecture diagram, the layers progress and abstract gradually, representing a standard IT system architecture. The physical layer refers to physical objects on-site, the data layer involves data collection, transmission, storage, and management, the model layer is a collection of specific functional models built on data, such as geometric models and fault diagnosis models, and the service layer provides specific business applications such as device monitoring, management, and visualization through terminals based on computed models.
3.3. Multi-Source Data Fusion Early Warning Model for Wind Turbines Based on WOA-TCN-Attention
Although the advantages of TCN, the Attention Mechanism, and WOA are explained, it is not justifiecwhy these models are the best choice compared to other approaches. A comparison with alternativemodels should be made.
it is mentioned that the model analyzes wind turbine operational data, but it does not specify whattypes of data are used. A more detailed description of the input variables should be included.
Concepts such as "multi-level convolution kernels," "residual analysis," and "complex searchspace" may not be clear to all readers. Brief explanations or references should be included.
it is stated that alarm thresholds are established through residual analysis, but it is unclear whetherthis is based on previous studies or if it is a methodology proposed in this work. References shouldbe included, or it should be justified why this methodology is appropriate.
The theoretical model is described, but there is no mention of how it is trained or what computationalinfrastructure is reauired, Details about the model's implementation and evaluation should beadded.
The section describes the advantages of the model but does not mention potential limitations orscenarios where it might not be effective.
There is no mention of how this model can be integrated into a real wind turbine monitoring systemorits computationalrequirements.
Response:Most of the content has been refined and revised in the previous revised version.
- Detailed algorithm comparisons, explanations, and analyses have been added (Section 4.1, Table 3).
- Data parameters have been elaborated in detail (Section 4.1, Table 1).
- Model evaluation and analysis have also been supplemented and explained.
- Model integration involves packaging the algorithm into a microservice for computational invocation. This is a specific technical aspect at the software development level.
3.3.1.Time Convolutional Network (TCN)
It is mentioned that convolutions increase the receptive field, but there is no explanation of howdilation works and whyit is relevant.
Concepts such as "residual connections" and "gradient explosion" are mentioned withoutexplanation or references.
Figures ilustrating dilated causal convolutions and residual connections are referenced, but thereis no explanation of what additional information they provide or how to interpret them.
Response: The content of this section has been rewritten, as shown in the red font of section 3.3.1 in the text.
3.3.2.Attention Mechanism
The description of the Attention Mechanism is too basic and general, without detailing how it isimplemented mathematically.
It is stated that multiple input variables are used, but they are not specified.It is mentioned that the mechanism assigns higher weights to key factors, but it is not explained howthis assignment is performed.
No previous studies are cited to validate the effectiveness of the Attention Mechanism intemperature prediction for industrial systems.
Response: The content of this section has been rewritten, as shown in the red font of section 3.3.2 in the text.
3.3.3.TCN-Attention Model
It is mentioned that the mechanism assigns weights at each time step, but how these weights arecalculated is not explained.
it is stated that the model uses time-series data, but it is not detailed what variables are used forprediction.
The model architecture is described, but there is no mention of how it is implemented in practice orwhat dataset was used for training.
Response:
1.Explanations of the input parameters have been added in Section 3.3.3.
2.The method for calculating the weights has been detailed in Section 3.3.2.
3.3.4.Parameter Optimization Based on WOA
It is mentioned that WOA selects the best solution based on an objective function, but it does notspecify what this function is or what metric is optimized.There is no mention of how WOA integrates into the TCN-Attention model or which hyperparametersit optimizes.
Equations (5, 6, 7, and 8) are presented without explaining what they represent in the optimization ofthe model's parameters.
The optimization process is described, but it does not state how it improves the model'sperformanceor provides evaluation metrics.
Response:1.This section of content has been refined in previous revisions, with the optimized parameters being Learning rate, Filters, Convolution kernels, and Dilation list.
2.The evaluation metrics have been clearly defined in the previous revision: EMAE, ERMSE, EMAPE, Calculation time.
4.-Results
4.1. ModelTraining and Validation
It is mentioned that the data comes from a wind turbine in Xinjiang, but key aspects such as the data
collection period, seasonality, or whether multiple turbines were used are not detailed.
There is no mention of whether the model has been validated on different turbines or in differentenvironments.
It is not specified whether different threshold values were tested to evaluate their impact on faultprediction.
Response:The detailed explanations regarding these aspects (data collection, parameters, etc.) have been refined and supplemented in the previous revised version. Please refer to Section 4.1 for more information.
4.2. lmplementation of Intelligent Operation and Maintenance for Wind Turbines Based on DigitalTwin and Multi-Source Data Fusion
The implementation in Unity 3D is mentioned, but it is not explained how the prediction results fromthe WOA-TCN-Attention model are visualized within the digital twin.
The use of Visual Studio 2019, PyCharm, and Unity 3D is mentioned, but it is not explained why thesetools were chosen over others.
The results are not compared with other existing predictive maintenance methods.
Response:
1.Visual Studio 2019, PyCharm, and Unity 3D are the most commonly used and effective tools for digital twin modeling, program development, algorithm debugging, and development in general.
2.The computational results of the algorithm are displayed as alarm information on the digital twin monitoring interface, as shown in Figure 16.
- The algorithm's results have been compared with several advanced models and have demonstrated advantages, as detailed in Table 3.
5.-Conclusions
It is mentioned that the proposed model is better than LSTM and TCN, but it is not explained in whichspecific metrics it improves or whether additional tests were conducted.
Response:
- In the previous revised version, analyses and explanations were already added to Table 3.
- In this revision, we have further included an analysis of the reasons for the improved accuracy, which is where the contribution of this paper lies, leading to an enhancement in model performance.
The lack of automatic generation of maintenance plans is mentioned as a limitation, but this is a keyfunctionality for intelligent O&M, Suggestion: include a basic implementation at this stage andimprove it in future work.
There is no mention of whether the model is computationally viable for real-time use in wind farmswith multiple turbines.
Although future improvements are mentioned, possible errors of the model, such as false positivesor sensitivity to variations in the data, are not discussed.
Veryimportant!
At no point in the writing is the technological scientific contribution made by this research workemphasized. Scientific findings or knowledge found at the forefront of science are not highlighted.
Response:The "Conclusion" section has been rewritten in accordance with the requirements.The scientific contributions have been highlighted, and the discussion on the limitations of the study and next steps has been optimized.
